# Intrinsic bias estimation for improved analysis of bulk and single-cell chromatin accessibility profiles using SELMA

Shengen Shawn Hu [1], Lin Liu[2], Qi Li[3], Wenjing Ma [1,4], Michael J. Guertin[5], Clifford A. Meyer [6,7], Ke Deng [3], Tingting Zhang [8] & Chongzhi Zang [1,9,10] ✉

Genome-wide profiling of chromatin accessibility by DNase-seq or ATAC-seq has been widely used to identify regulatory DNA elements and transcription factor binding sites. However, enzymatic DNA cleavage exhibits intrinsic sequence biases that confound chromatin accessibility profiling data analysis. Existing computational tools are limited in their ability to account for such intrinsic biases and not designed for analyzing single-cell data. Here, we present Simplex Encoded Linear Model for Accessible Chromatin (SELMA), a computational method for systematic estimation of intrinsic cleavage biases from genomic chromatin accessibility profiling data. We demonstrate that SELMA yields accurate and robust bias estimation from both bulk and single-cell DNase-seq and ATAC-seq data. SELMA can utilize internal mitochondrial DNA data to improve bias estimation. We show that transcription factor binding inference from DNase footprints can be improved by incorporating estimated biases using SELMA. Furthermore, we show strong effects of intrinsic biases in single-cell ATAC-seq data, and develop the first single-cell ATAC-seq intrinsic bias correction model to improve cell clustering. SELMA can enhance the performance of existing bioinformatics tools and improve the analysis of both bulk and single-cell chromatin accessibility sequencing data.

Cis-regulatory elements in the genome play a critical role in transcription regulation by interaction with protein molecules such as transcription factors (TFs). These DNA elements are usually unwrapped from packed nucleosomes and are accessible in the chromatin structure[1,2]. Genome-wide profiles of chromatin accessibility are a means to measure the global landscapes of active regulatory elements in different cell types. DNaseI hypersensitivity sequencing (DNase-seq) and the assay for transposase-accessible chromatin using sequencing

(ATAC-seq) have become widely used for the genomic profiling of chromatin structure and accessibility[3,4]. Signal enrichments, or "peaks", from DNase-seq or ATAC-seq data are considered to represent accessible chromatin regions and can be used for inferring regulatory elements or TF binding sites. In addition, DNase-seq and ATAC-seq data also exhibit footprint patterns, which are fine structures in the accessible chromatin where high-affinity protein-DNA interactions protect the DNA from DNaseI or Tn5-transposase cleavages[4,5].

[1]Center for Public Health Genomics, University of Virginia, Charlottesville, VA 22908, USA. [2]Institute of Natural Sciences, MOE-LSC, School of Mathematical Sciences, CMA-Shanghai, and SJTU-Yale Joint Center for Biostatistics and Data Science, Shanghai Jiao Tong University, Shanghai 200240, China. [3]Center for Statistical Science and Department of Industrial Engineering, Tsinghua University, Beijing 100084, China. [4]Department of Computer Science, Emory University, Atlanta, GA 30322, USA. [5]Center for Cell Analysis and Modeling, Department of Genetics and Genome Sciences, University of Connecticut, Farmington, CT 06030, USA. [6]Department of Data Science, Dana-Farber Cancer Institute, Boston, MA 02215, USA. [7]Department of Biostatistics, Harvard T.H. Chan School of Public Health, Boston, MA 02115, USA. [8]Department of Statistics, University of Pittsburgh, Pittsburgh, PA 15260, USA. [9]Department of Public Health Sciences, University of Virginia, Charlottesville, VA 22908, USA. [10]UVA Comprehensive Cancer Center, University of Virginia, Charlottesville, VA 22908, USA. ✉e-mail: zang@virginia.edu

DNase/ATAC-seq footprint detection has been implicated as an effective approach for identifying accurate TF binding sites at base-pair resolution[6,7]. A few computational tools have been developed for detecting footprints from DNase-seq (RepFootprint[8], Wellington[9], PIQ[10]) or ATAC-seq data (HINT-ATAC[11], ToBIAS[12]). A recent study integrated 243 DNase-seq samples from different human cell and tissue types and reported approximately 4.5 million DNaseI consensus footprints associated with TF occupancy across the human genome as one of the largest maps of human regulatory DNA[7].

The premise of using DNase-seq and ATAC-seq data to profile chromatin accessibility is that enzymatic DNA cleavages represented by sequence reads reflect local chromatin openness only. However, it has been shown that both DNaseI and Tn5 transposase exhibit sequence selection biases in high-throughput sequencing data[13-16]. Such intrinsic enzymatic biases in sequencing data can potentially confound observed cleavage patterns and footprint detection. The characterization and correction of intrinsic cleavage biases are essential to DNase/ATAC-seq data analysis. To characterize intrinsic cleavage biases, most studies use a k-mer model, in which the k-mer DNA sequence centered at a cleavage site of DNaseI/Tn5 is used as the signature of this cleavage[13,16,17]. The sequence bias can be estimated by counting the occurrences of cleavages with each k-mer in one dataset relative to the genome-wide occurrences of this k-mer. A naive k-mer model assumes that k-mers are independent of each other, resulting in an exponential increase in the degree of freedom when k increases. Therefore, a naive k-mer model becomes less feasible in practice with a large k, especially with samples having insufficient sequencing depth. Although most studies use 6-mers with $4^6 = 4096$ parameters[8,10,13-18], it is unclear whether a different model with a larger k-mer size and more feasible parameter estimation can achieve better performance.

Several studies have used various computational approaches for intrinsic cleavage bias estimation[8,14,16,19] and footprint detection with bias correction[16,18-20] for bulk DNase/ATAC-seq data. Recently, single-cell ATAC-seq (scATAC-seq) has enabled chromatin accessibility profiling in thousands of individual cells at one time[21-24]. Due to the high sparsity of single-cell data and because most chromatin accessibility regions contain only one read in one cell, any potential bias can be substantial in scATAC-seq data compared to bulk data, creating additional challenges in computational analysis. Incorporating intrinsic cleavage bias effects for improved scATAC-seq analysis also requires comprehensive assessment and development of innovative methods.

Here, we present Simplex Encoded Linear Model for Accessible Chromatin (SELMA), a computational framework for the accurate estimation of intrinsic cleavage biases and improved analysis of DNase/ATAC-seq data for both bulk and single-cell experiments. We demonstrate that SELMA generates more accurate and robust bias estimation from bulk DNase/ATAC-seq data than the naive k-mer model and that SELMA can utilize mitochondrial DNA (mtDNA) for bias estimation instead of requiring a separate naked DNA sample. We show an improved TF occupancy inference on ENCODE consensus footprints by including SELMA-estimated biases for each footprint. Finally, we show that SELMA-estimated biases can be incorporated with existing scATAC-seq computational tools to generate more accurate cell clustering analysis.

## Results

### SELMA improves cleavage bias estimation in DNase-seq and ATAC-seq data

We developed SELMA for an accurate and robust estimation of intrinsic cleavage biases from chromatin accessibility sequencing data. In SELMA, we start with a naive k-mer model to calculate the frequency of observed cleavage sites at each k-mer relative to the total occurrences of this k-mer (Fig. 1a). We further encode each k-mer as a vector in the Hadamard Matrix $H_{16}$, derived from a simplex encoding model, in which the k-mer sequences are encoded as the

vertices of a regular 0-centered simplex[25,26]. Intuitively, a k-mer can be represented as k mononucleotides and $k-1$ adjacent dinucleotides. Each mononucleotide is encoded as the 3-dimensional vector of one of the four tetrahedral vertices of the cube of side 2 centered at the origin. Each dinucleotide is encoded as the outer product of the 2 vectors representing the associated nucleotides (Fig. 1b). Including an intercept, this k-mer simplex encoding model has a total of $1 + 3 \times k + 9 \times (k-1) = 12k - 8$ parameters, much fewer than the naive k-mer model ($4^k$). We use a linear model to fit these $12k - 8$ parameters from the naive k-mer biases, and use the fitted values as the SELMA-modeled cleavage biases (Fig. 1c).

As an intrinsic property of the enzyme (DNaseI/Tn5 transposase), the cleavage biases are expected to be invariant across cell types and independent of chromatin states (Supplementary Fig. 1a). Comparing data from two cell types using different 8-mer models as an example, we found that the biases estimated using SELMA have a higher correlation than those estimated using the naive k-mer model, for both DNase-seq (Fig. 1d, e) and ATAC-seq (Fig. 1f, g). Using sequence reads from genomic regions with different chromatin accessibility levels, the naive k-mer model was not able to generate very consistent bias estimations (Supplementary Fig. 1b, c), but the k-mer biases estimated by SELMA using the same data were highly consistent (Supplementary Fig. 1b-e). The sequence preferences of DNaseI or Tn5 cleavage can be better reflected when the enzymes are applied to deproteinized naked genomic DNA[16,17]. We found that the k-mer cleavage biases in naked DNA DNase/ATAC-seq data estimated with the naive k-mer model can still be improved by SELMA, obtaining more consistent bias scores between different cell systems (Fig. 1h-k). These data demonstrated that SELMA can improve the accuracy of estimating k-mer cleavage biases in DNase-seq and ATAC-seq data.

With fewer parameters, SELMA enabled us to assess the effect of k-mer size on the performance of bias estimation. Using a "bias-expected cleavage" approach[8,13,16,19], we compared the bias estimation performances measured by the correlation coefficient between the genome-wide observed cleavages and bias-expected cleavages estimated using SELMA with different k. A higher correlation coefficient indicates a more accurate bias estimation to calculate the expected cleavages. For both DNase-seq and ATAC-seq data from two different cell lines, we found that 10-mer outperforms any other k-mer (Fig. 1l-o). We then applied this analysis to more DNase-seq and ATAC-seq data from a variety of human tissues generated by ENCODE and found that 10-mer always exhibited the best performance (Supplementary Fig. 2). The above empirical evidence suggested that 10-mer is the optimal choice for intrinsic cleavage bias estimation for both DNaseI (DNase-seq) and Tn5 (ATAC-seq) cleavages.

### SELMA improves ATAC-seq bias estimation by considering dimeric Tn5 cleavages

Many studies processed DNase-seq and ATAC-seq data in a similar way, treating individual DNA cleavage sites directly as indications of accessible chromatin[8,14]. However, the mechanisms of enzymatic DNA cleavage are different between DNaseI and Tn5 transposases. Unlike DNaseI, the Tn5 transposase binds DNA as a dimer and inserts adapters on the two strands separated by 9 bp[4,27] (Fig. 2a). As a result, the presence of each observed Tn5 cleavage in ATAC-seq data should be the consequence of two insertion events induced by the same Tn5 dimer, and the bias estimation of a Tn5 cleavage site should consider both the observed cleavage and the coupling cleavage 9 bp downstream on the reverse strand. Therefore, SELMA estimated the bias on an ATAC-seq cleavage site as the geometric mean of the bias scores of the 10-mers at the 5' observed cleavage and at the 3' cleavage 9 bp downstream on the reverse strand (Fig. 2a). To show the dimeric Tn5 cleavage effect, we calculated the cross-correlation between the genome-wide profiles of the plus strand cleavages and the minus strand cleavages. As expected, we observed a peak at 9 bp of the cross-correlation curve specifically in

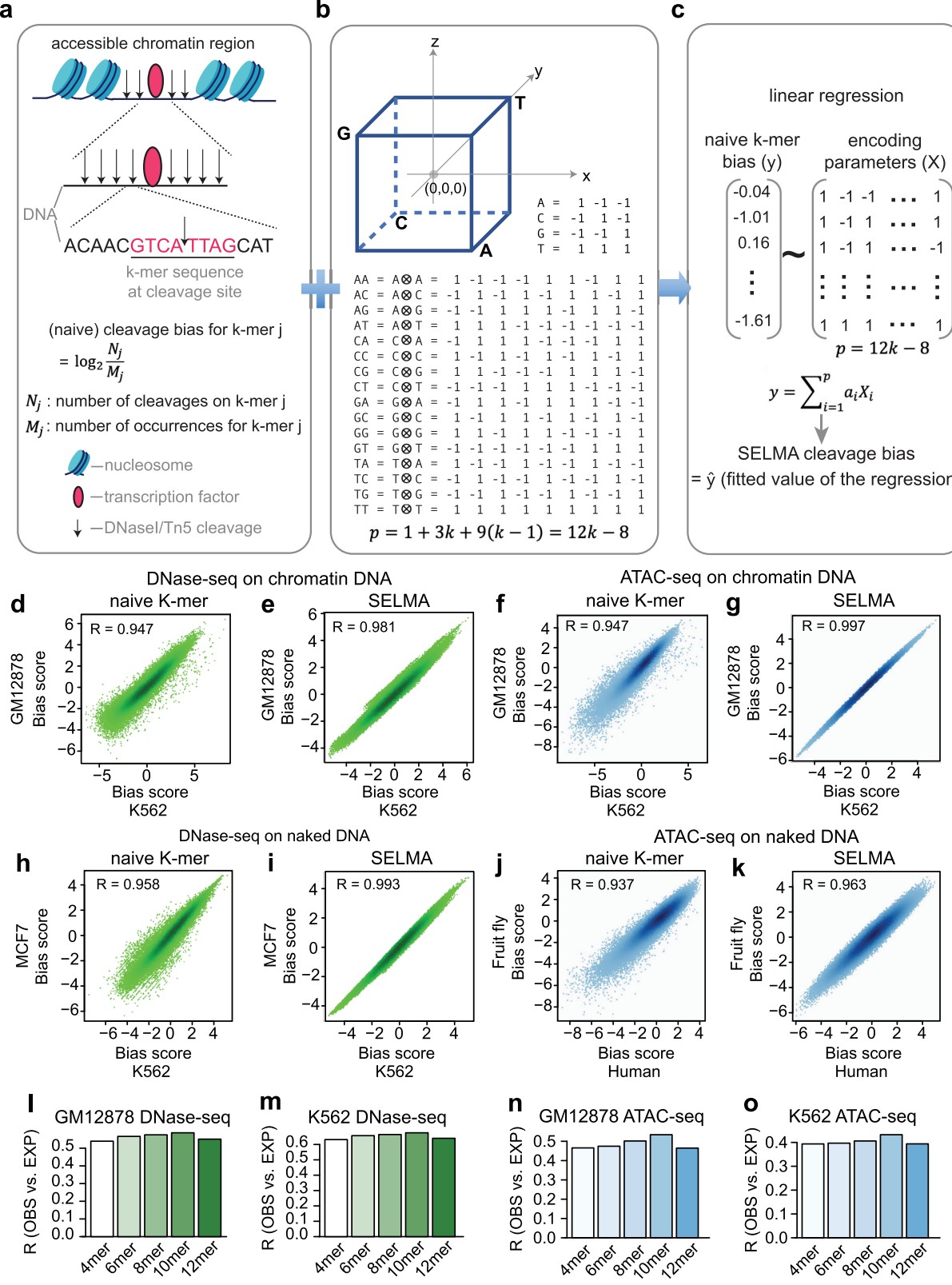

the ATAC-seq data but not the DNase-seq data (Fig. 2b, c). Similarly, we observed perfectly matching aggregate cleavage patterns on TF motif consensus sites between plus strand and minus strand cleavages shifted by 9bp (Supplementary Fig. 3a, b).

Using the similar "observed-expected correlation" approach described above, we found that SELMA considering dimeric cleavages outperformed models considering only 5′ cleavage in generating a more accurate bias-expected cleavage pattern for ATAC-seq data (Fig. 2d, e). We confirmed that this observation was specific to ATAC-seq, as similar analyses for DNase-seq from the same cell lines did not show a similar level of improvement (Supplementary Fig. 3c, d). We also compared SELMA with several existing bias estimation

**Fig. 1 | SELMA framework for cleavage bias estimation. a** Schematic of a naive k-mer model for cleavage bias estimation. **b** Simplex encoding model. The coordinates of the 4 tetrahedral vertices of the cube encode the 4 nucleotides. Each dinucleotide is encoded as the outer product of the 2 mononucleotides. *p* represents the number of parameters in a k-mer simplex encoding model. **c** SELMA uses k-mer simplex encoding followed by a linear regression for k-mer cleavage bias estimation. Comparison between the naive k-mer model (**d**, **f**, **h**, **j**) and SELMA (**e**, **g**, **i**, **k**) on k-mer cleavage bias scores estimated from DNase-seq data (**d**, **e**, **h**, **i**) and ATAC-seq data (**f**, **g**, **j**, **k**). Each dot in a scatter plot represents an 8-mer, with its estimated bias score from different datasets represented in the x- and y-axes. Chromatin DNA (**d–g**) and naked DNA (**h–k**) from different human cell lines (**d–i**) or different species (**j**, **k**) were compared as labeled. R represents the Pearson correlation coefficient. **l–o** Intrinsic cleavage bias estimation accuracy measured by correlation between genome-wide observed (OBS) and bias-expected (EXP) cleavages with different k-mers. A higher Pearson correlation coefficient (R) indicates a better prediction of the observed cleavage profile using the estimated biases.

approaches[18] and found that SELMA's performance was the best for ATAC-seq data from several cell lines as well as different human tissue types from ENCODE (Fig. 2d, e, Supplementary Fig. 3e–j). Several new Tn5-based techniques have recently been developed for improved chromatin accessibility profiling, including THS-seq[28], Fast-ATAC-seq[29], and Omni-ATAC-seq[30]. Data from these technologies showed intrinsic cleavage biases similar to those of conventional ATAC-seq (Supplementary Fig. 3k–n). We found that SELMA also outperformed other approaches in bias estimation (Fig. 2f–i). Collectively, these data suggested that SELMA can most accurately estimate intrinsic cleavage biases in data from ATAC-seq and other Tn5-based techniques.

## SELMA enables accurate bias estimation by utilizing mitochondrial DNA (mtDNA) reads

Accurate estimation of enzymatic cleavage biases independent of chromatin usually requires a control sample of deproteinized naked DNA digested by the same enzyme. Biases estimated from the naked DNA control data can be used to correct the chromatin accessibility profiling data[13,14,16–18] (Fig. 3a). In conventional DNase/ATAC-seq data analyses, sequence reads mapped to mitochondrial DNA (mtDNA) are usually discarded[4]. Lacking histones responsible for the chromatin structure, mtDNA is similar to deproteinized naked DNA[31–33]. Therefore, we sought to use the mtDNA reads from DNase/ATAC-seq data for cleavage bias estimation as an alternative to using a naked DNA control sample (Fig. 3b). Likely because of relatively low read counts and lack of sequence diversity (e.g., human mtDNA is only <20 kb long), the naive k-mer model exhibited poor performance on bias estimation from mtDNA reads, using bias scores estimated from naked DNA as a reference (Fig. 3c). In contrast, SELMA generated a more accurate bias estimation from the same mtDNA reads, which was highly consistent with the bias scores estimated from the naked DNA data (Fig. 3d), demonstrating the power of SELMA to use less input to make accurate bias estimations. We applied this approach to a series of ATAC-seq and DNase-seq datasets for different human tissues from ENCODE and found that SELMA was better than the naive k-mer model in yielding a consistently higher correlation coefficient (>0.9) between mtDNA-estimated bias and naked DNA-estimated bias for every ATAC-seq and DNase-seq sample tested (Supplementary Fig. 4a, b). Many optimized ATAC-seq protocols aimed to reduce mtDNA reads to increase the fraction of chromatin DNA reads for chromatin accessibility signal yield[29,30]. We sampled down mtDNA reads to test the performance of SELMA in making robust bias estimations and found that SELMA could accurately estimate the bias with as few as 50,000 mtDNA reads (Fig. 3e, Supplementary Fig. 4c), which is approximately 0.2% of the sequencing depth of a regular ATAC-seq sample and lower than the fraction of mtDNA reads in any existing ATAC-seq experiment[29,30]. These data demonstrated that by applying SELMA, mtDNA reads can be utilized to substitute naked DNA control samples for accurate bias estimation.

## Considering SELMA-estimated bias improves TF binding inference on ENCODE DNaseI footprint regions

With an accurate bias estimation model developed, we next sought to examine the potential effect of intrinsic biases on TF binding footprints in chromatin accessibility profiling data. Focusing on the 4.5 million human DNaseI consensus footprints across the human genome, we plotted the DNaseI cleavages from different human cell lines and observed similar classic DNaseI cleavage protection ("footprint") patterns across these footprint regions (Fig. 4a, Supplementary Fig. 5a). Interestingly, we also observed a recurrent pattern of the SELMA-estimated DNaseI cleavage bias that is well aligned with the DNaseI cleavage pattern across these footprint regions (Fig. 4b, Supplementary Fig. 5b). In the aggregate view of footprint regions of different lengths, the DNaseI cleavage signals exhibited a clear "cliff-bound valley"-shaped footprint pattern (Supplementary Fig. 5c). The DNaseI cleavage bias scores exhibited a pattern of two spikes located inside the footprint coordinates (Supplementary Fig. 5d). After normalizing various footprint lengths, we plotted the aggregate DNaseI cleavage patterns across all consensus footprint regions and found that the overall "footprint pattern" clearly included the double spike pattern of cleavage biases (Fig. 4c), which aligned well with the two spikes in the aggregate bias score pattern (Fig. 4d), indicating a possible contribution of intrinsic biases to the DNaseI footprinting.

To assess the interference of intrinsic biases with DNaseI cleavage patterns at TF binding footprints, we aligned the 4.5 million consensus footprints with more than 10,000 human TF ChIP-seq datasets from the Cistrome Data Browser database[34,35] and collected two sets of footprint regions: "TF binding hotspots", the footprint regions overlapping with TF binding sites detected from more than 3000 ChIP-seq datasets, and "TF binding deserts", the footprint regions that do not overlap with any TF binding sites from any ChIP-seq dataset or any human TF motif sequence from the HOCOMOCO database[36] (Supplementary Fig. 5e). We compared the aggregate DNaseI cleavage patterns and the bias score patterns in these two sets of consensus footprint regions and found that the DNaseI cleavage pattern in TF binding hotspot regions was dominated by the classic DNaseI footprint pattern with little interference from the bias (Fig. 4e, f), while in the TF binding desert regions, the entire cleavage "footprint" pattern was apparently determined by the two spikes from the intrinsic bias (Fig. 4g, h). These results suggested that although the overall DNaseI cleavage pattern is indicative of TF binding, the intrinsic cleavage bias may interfere with the real footprint pattern, and the effect on those footprint regions with rare TF binding events can be drastic. These observations were reproducible in DNase-seq data from different cell and tissue types (Supplementary Fig. 6a, b).

To quantify the level of intrinsic cleavage biases in a consensus footprint region, we defined a footprint bias score (FBS) as the relative SELMA-estimated bias score at the two spikes compared to the average SELMA bias score across the rest of the region for each footprint. Consistent with what we observed in the aggregate view, the FBSs for TF binding deserts were significantly higher, while the FBSs for TF binding hotspots were significantly lower, than the background of all consensus footprints (Fig. 4i, $p < 10^{-5}$, by one-sided Wilcoxon rank-sum test), indicating that FBS might be an informative feature of the consensus footprint regions and might help separate true TF binding footprints from false-positive DNaseI footprint patterns. Next, we used a model-based approach to assess the potential power of SELMA-derived FBS in boosting the performance of TF binding inference from DNase-seq signal patterns on consensus footprint regions containing the TF motif sequence. For every TF with a known motif in a cell type with both ChIP-seq and DNase-seq data available, we employed a multinomial logistic regression model using different sets of features,

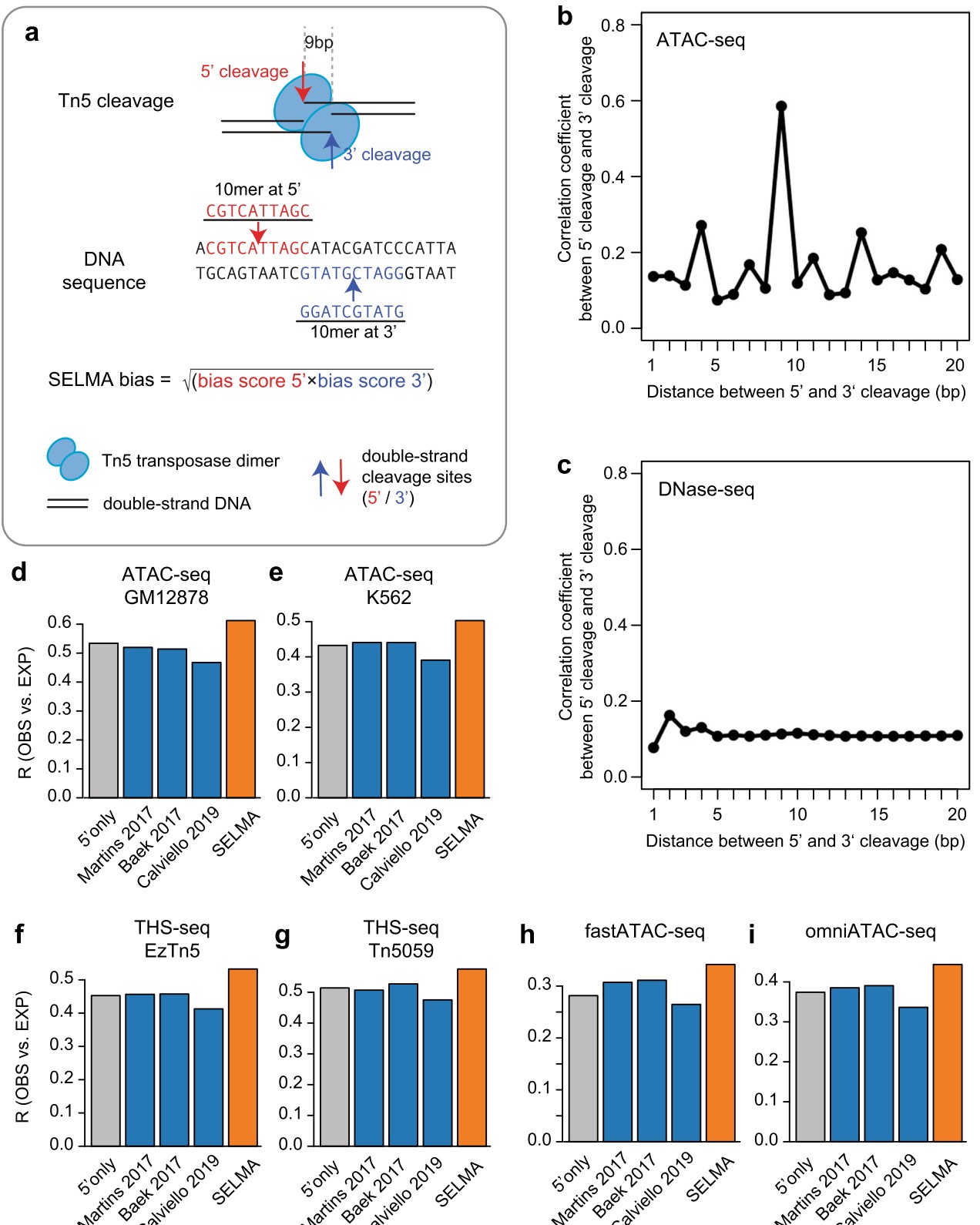

**Fig. 2 | SELMA consideration of dimeric Tn5 cleavages for ATAC-seq.**
**a** Schematic of SELMA consideration of dimeric Tn5 cleavages. **b**, **c** Cross-correlation between 5′ cleavages and 3′ cleavages across genome-wide accessible chromatin regions in the human GM12878 cell line from ATAC-seq (**b**) and DNase-seq (**c**) experiments. The x-axis represents the shift distance (in bp) between 5′ and 3′ cleavages. **d**–**i** Comparison of ATAC-seq intrinsic cleavage bias estimation accuracy measured by Pearson correlation coefficients (R) between genome-wide observed (OBS) and bias-expected (EXP) cleavages for human GM12878 (**d**) and K562 (**e**) cell lines, as well as several modified Tn5-based assays (**f**–**i**). Different bars represent different estimation approaches: gray for considering the 10-mer at the observed cut only (5′ only); orange for SELMA; and blue for several published approaches. Modified Tn5-based assays include **f** THS-seq with standard Tn5; **g** THS-seq with mutated Tn5; **h** fast-ATAC-seq; and **i** omni-ATAC-seq.

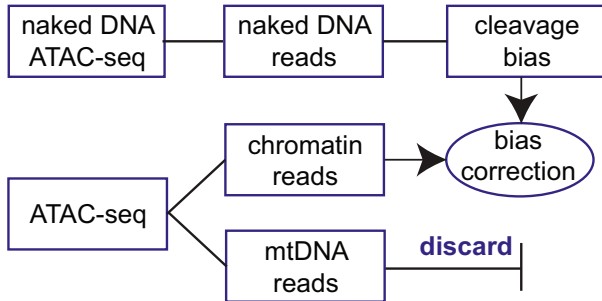

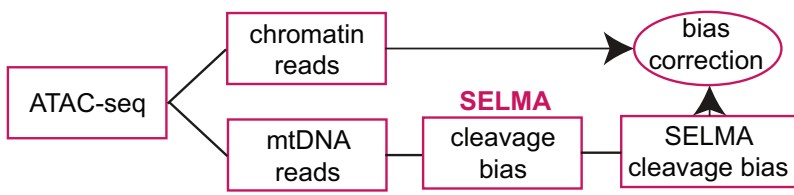

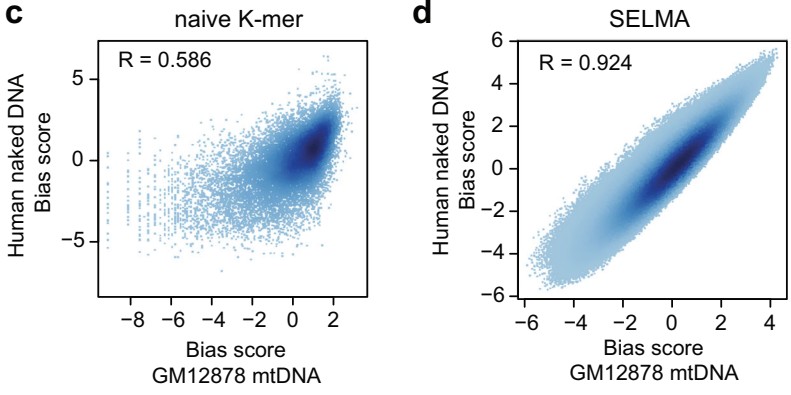

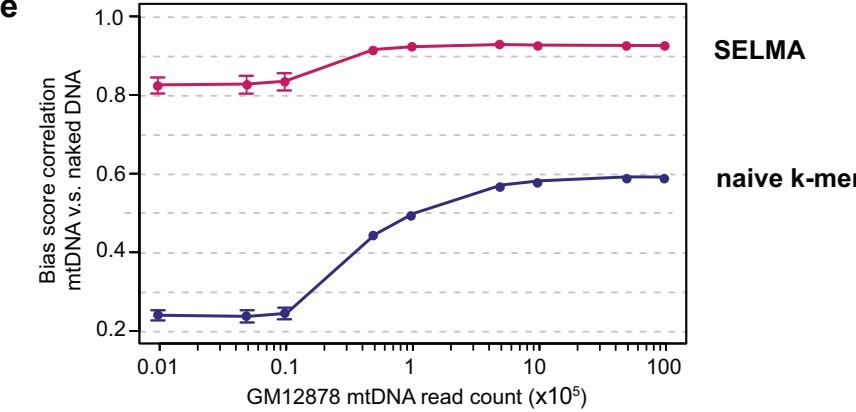

**Fig. 3 | SELMA bias estimation using mtDNA reads. a** Conventional approach of intrinsic cleavage bias estimation, in which a naked DNA sample is required for bias estimation, and mtDNA reads from the chromatin DNA ATAC-seq experiment are discarded. **b** SELMA approach of intrinsic cleavage bias estimation using mtDNA reads. Chromatin reads and mtDNA reads from the same ATAC-seq dataset were separated for Tn5 cleavage profiling and bias estimation with SELMA, respectively. **c**, **d** Scatter plots demonstrating the robustness of 10-mer bias scores estimated from mtDNA (mtDNA reads from chromatin ATAC-seq data, *x*-axis) and genomic DNA (reads from naked DNA ATAC-seq data, *y*-axis), comparing the naive k-mer model (**c**) and SELMA (**d**). SELMA shows a more robust estimation using mtDNA reads with higher correlation than the naive k-mer model. **e** Comparison of the correlation coefficients between biases estimated from mtDNA reads and from naked DNA with different mtDNA read counts from down-sampling. Error bars represent the standard deviation from 10 runs with different random seeds ($n = 10$).

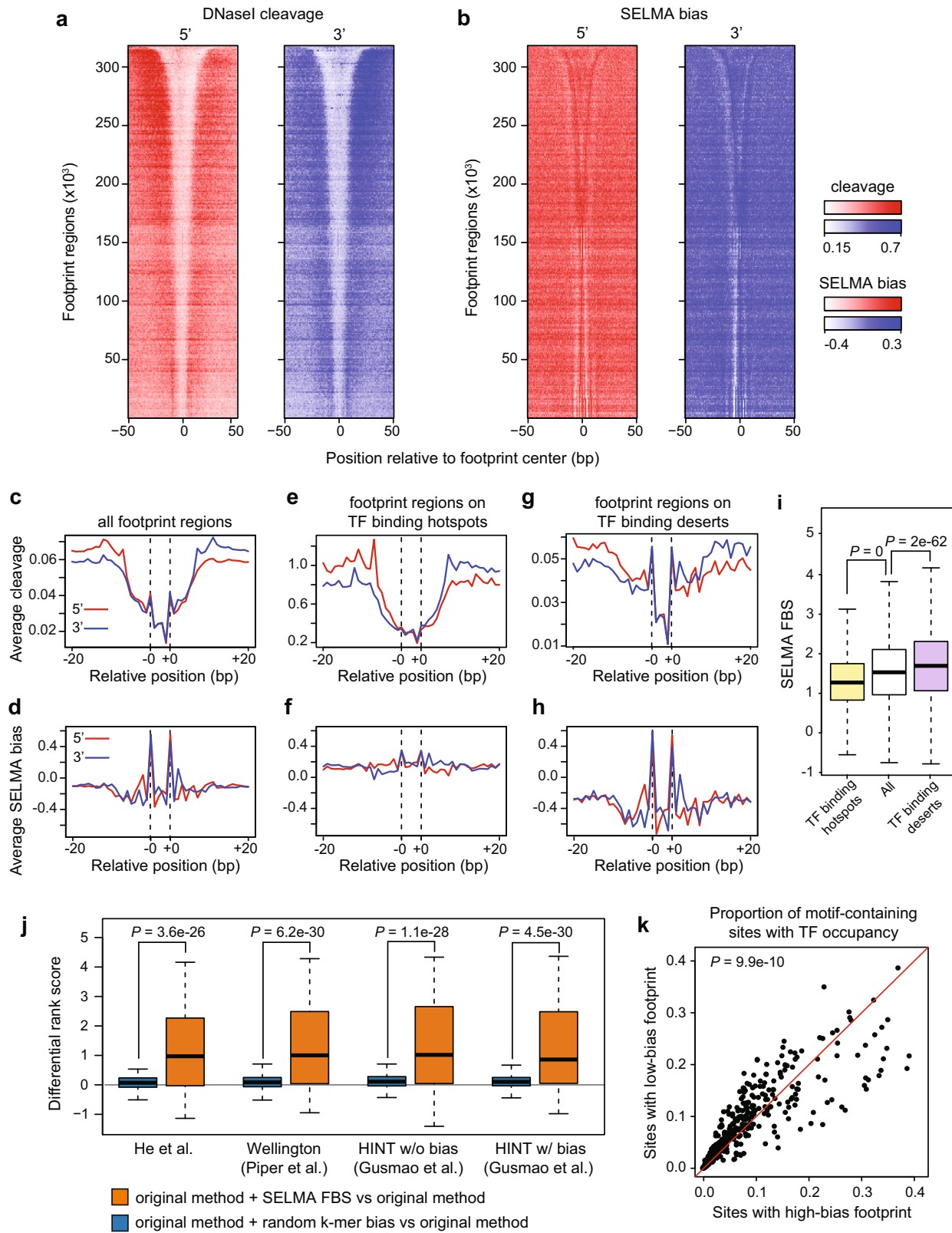

each of which may include DNase-seq read count, DNaseI footprint score produced from an existing method, and an optional FBS, to infer the TF binding occupancy (determined by peak occurrence in ChIP-seq data) in the motif-containing footprint regions. For each footprint method, we compared the TF binding inference performance of the original method (read count + footprint score as features), the original

method plus a randomized naive k-mer bias score feature as a control, and the original method plus the SELMA bias score (FBS), using an integrated rank score strategy. We included our previous footprint method[13] and several representative methods that outperform others, Wellington[9] and HINT[11,19] (with and without bias correction mode) in this comparison. We surveyed all human cell types that have both

**Fig. 4 | TF binding inference on DNaseI footprint regions is affected by intrinsic cleavage biases and improved by SELMA.** Heatmaps of DNaseI cleavage patterns in the GM12878 cell line (**a**) and SELMA-estimated bias scores (**b**) around ENCODE DNaseI consensus footprint regions. 5′ and 3′ patterns were plotted separately. Footprint regions were consistently ranked by footprint length, and signals for every 1000 regions with similar lengths were averaged as one row in the heatmap. Aggregate plots of DNaseI cleavage patterns (**c, e, g**) and SELMA-estimated bias scores (**d, f, h**) at all 4,460,438 ENCODE DNaseI consensus footprint regions (**c, d**), 40,110 footprint regions overlapping with TF binding hotspots (**e, f**), and 10,106 footprint regions overlapping with TF binding deserts (**g, h**). DNaseI cleavages are from GM12878 (**c** and **e**) and merged from GM12878, K562 and ENCODE tissues (**g**). Dashed lines represent estimated bias spikes in the footprint regions. **i** SELMA-estimated footprint bias scores (FBS) for footprint regions at TF binding hotspots (yellow; $n = 40,110$), all footprint regions (white; $n = 4,460,438$), and footprint regions at TF binding deserts (purple; $n = 10,106$). **j** Difference of performance rank scores for transcription factor binding inference from DNaseI footprint using various methods for human cell lines ($n = 375$ TFs for each boxplot). Boxplots with different colors represent different approaches as indicated in the legends. **k** Scatter plot showing the heterogeneous effect of intrinsic cleavage biases on different TF motifs. Each data point represents motif sites for one TF in one cell type. Fractions of the TF-bound motif sites overlapping with high-FBS footprint regions and with low-FBS footprint regions are plotted on the x- and y-axes, respectively. More motifs located above the diagonal line indicate that more TFs are more likely to bind at low-FBS sites than at high-FBS sites. The p value was calculated by one-tailed t-test comparing the distribution of log-likelihood ratios (y-axis/x-axis) to the standard normal distribution. The centerline, bounds of box, top line, and bottom line of the boxplots represent the median, 25th to 75th percentile range, 25th percentile − 1.5 × interquartile range (IQR), and 75th percentile + 1.5 × IQR, respectively. All P values were calculated by the one-sided Wilcoxon rank-sum test.

DNase-seq and more than 20 TF ChIP-seq samples available from ENCODE, including in 7 cell lines and a total of 375 ChIP-seq samples for 156 different TFs (Supplementary Dataset 1, Supplementary Fig. 7a). For each TF ChIP-seq sample in each cell line, we calculated differential rank score of the inference performance by adding a bias score feature. As a result, adding a random k-mer bias score did not change the inference performance. In contrast, the models with SELMA FBS added can significantly increase the rank scores for the majority of ChIP-seq samples, regardless of which method was used originally to calculate the footprint score (277–291, or 74–78%, varying across different footprint methods, Fig. 4j, Supplementary Fig. 7, Supplementary Dataset 2). For example, using our previous footprint method[13], 277 ChIP-seq samples (74%) received a higher inference rank score when considering SELMA FBS, covering 117 (75%) of the 156 TFs (Supplementary Dataset 2). Among these, Zinc finger family TFs including CTCF and REST showed the highest improvement after considering footprint bias (Supplementary Fig. 8, Supplementary Dataset 2), consistent with previous studies about the positive correlation between footprint strength and residence time of the TF on DNA[13,16]. Meanwhile, the SOX family (e.g., SOX5) and HLH family (e.g., MYC) TFs rarely showed improved inference performance after considering footprint bias (Supplementary Fig. 8, Supplementary Dataset 3). These TFs were those having short residence time on DNA and weak footprints[16], which were expected not to be affected by footprint biases. These results demonstrated that considering the intrinsic cleavage bias can improve the performance of existing footprint computational methods for the binding inference for most TFs with footprints.

To assess whether the intrinsic bias has different levels of interference with the footprint regions for different TFs, we selected two subsets of sequence motif-containing footprint regions for each TF as the top 10% with the highest FBS or the bottom 10% with the lowest FBS, and compared the frequencies of observing actual TF binding (overlapping with a ChIP-seq peak) in these subsets of footprints. We found that different TFs had various preferences for binding at low-FBS footprints or high-FBS footprints. Among the 156 TFs included, a significantly larger amount of TFs can be improved with SELMA FBS than those that cannot be improved by SELMA (Fig. 4k). These results suggested that intrinsic cleavage biases might affect different TFs at various levels in divergent directions, and considering intrinsic bias should improve the footprint-based TF binding inference for most TFs.

### SELMA improves the accuracy of cell clustering from single-cell ATAC-seq data

Single-cell ATAC-seq (scATAC-seq) technologies enable the detection of accessible chromatin regions at single-cell resolution in thousands of cells at a time[21–23]. Due to the scarcity of cleavage events in an individual cell, most chromatin accessibility regions in a single cell have only one aligned fragment, making the potential influence of intrinsic cleavage biases more substantial in scATAC-seq data than in bulk ATAC-seq data. We collected scATAC-seq datasets generated using different platforms for different biological samples, human hematopoietic cells[37], mixed human cell lines[21], and mouse primitive gut tube[38], and found that the scATAC-seq data contained similar intrinsic cleavage biases to bulk data with highly correlated bias scores estimated by SELMA (Supplementary Fig. 9a–c). We estimated the average cleavage bias for each individual cell (cell bias score, CBS) and found that the distribution of CBS was different across cell cluster patterns, batches, and annotated cell types (Fig. 5a–i, Supplementary Dataset 4). We further surveyed datasets from the 10x Single-Cell Multiome platform for different biological systems including mouse embryonic brain, human peripheral blood mononuclear cells (PBMC), and human lymph node, each of which has scATAC-seq and scRNA-seq performed in the same cell and we used the scRNA-seq derived cell clusters as the "pseudo" ground truth to label the cells. As a result, we still found similar intrinsic cleavage biases in the scATAC-seq part of the data (Supplementary Fig. 9d–f) and CBS affecting different cell clusters (Fig. 5j–r, Supplementary Dataset 4). These observations indicated that, regardless of experimental platforms and biological systems, the Tn5 intrinsic cleavage biases can contribute to cellular heterogeneity observed from scATAC-seq data and can affect cell clustering analysis.

We next assessed whether considering intrinsic cleavage biases can improve cell clustering based on scATAC-seq data. We used the actual cell-type labels as the clustering ground truth for the human hematopoietic cell sample and the mixed human cell line sample, and used scRNA-seq data-projected cell labels as a "pseudo" ground truth for the other samples. We used the adjusted Rand index (ARI)[39] to quantify the accuracy of an unsupervised clustering result. As scATAC-seq-based cell clustering can be performed on all chromatin accessibility regions (ATAC-seq peaks), we sought to address whether removing peaks with high intrinsic biases can increase the clustering accuracy. We first tested a simple clustering approach that involved principal component analysis (PCA) dimensionality reduction followed by K-means clustering. For most cases in the 6 scATAC-seq datasets, cell clustering after removing 1–50% of peaks with the highest PBS can increase ARI from using all peaks (Supplementary Fig. 10). Although the level of improvements varies across different datasets and not every percentage of peak removal yields a higher ARI, such improvement in clustering accuracy by removing high-bias peaks is statistically significant compared to the control of randomly removing the same number of peaks (Fig. 6a–f). These data suggested that scATAC-seq cell clustering could be improved by excluding high-bias peaks that confounded the analysis.

To correct the intrinsic cleavage bias effect in scATAC-seq data in a systematic manner, we developed a general model to weight peaks by the percentile of SELMA PBS (Fig. 6g). The weight function was determined empirically using a Beta distribution based on the relative contribution of each PBS percentile of peaks to the true cell type

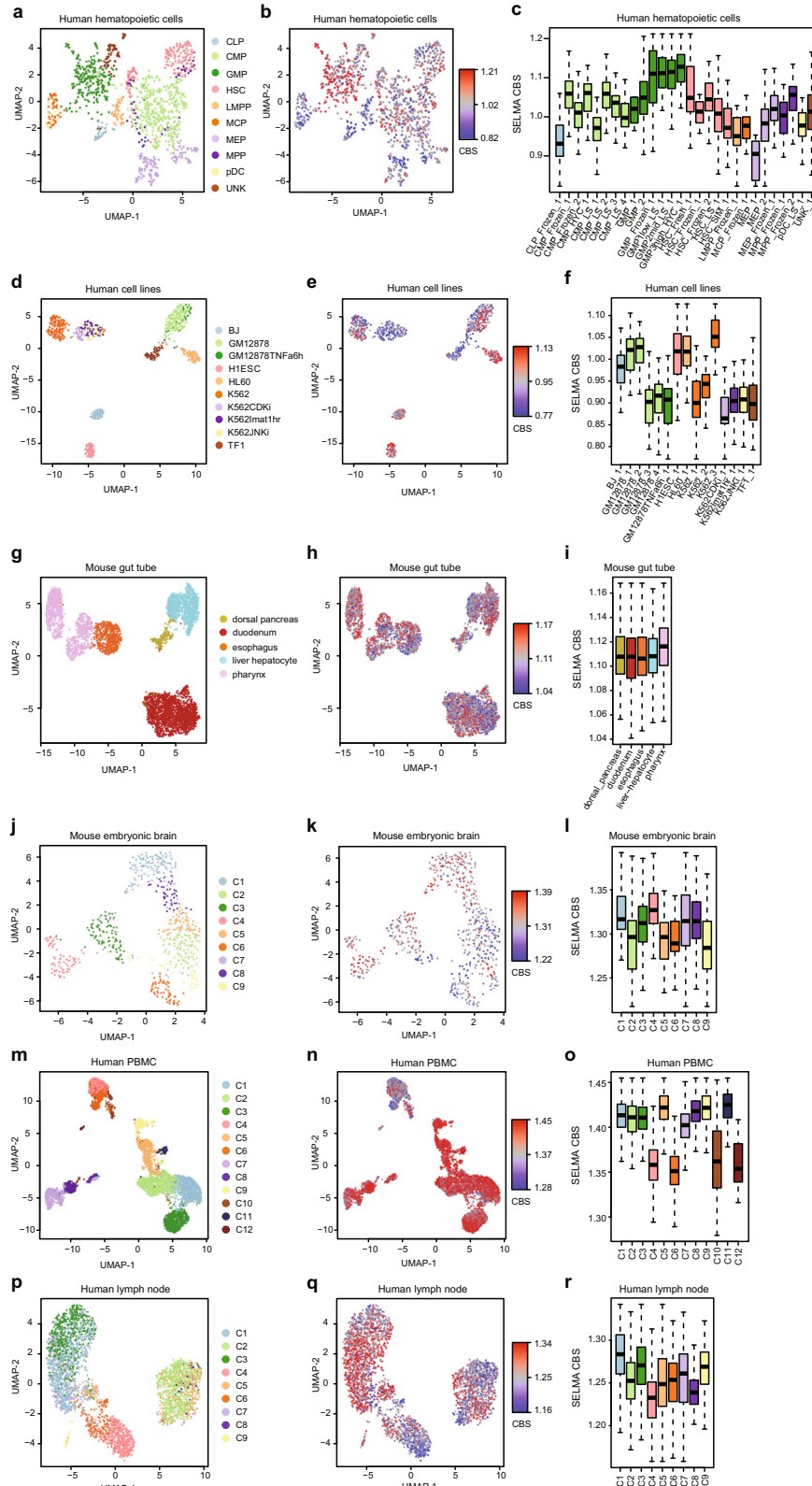

**Fig. 5 | Intrinsic cleavage biases affect single-cell ATAC-seq data analysis.**
Visualization of intrinsic cleavage bias effect in different cell clusters derived from scATAC-seq data for different biological samples and different experimental platforms: human hematopoietic cells (**a**–**c**), mixed human cell lines (**d**–**f**), mouse primitive gut tube (**g**–**i**), and 10× Single-Cell Multiome data for mouse embryonic brain (**j**–**l**), human peripheral blood mononuclear cells (PBMC) (**m**–**o**), and human lymph node (**p**–**r**). **a**, **d**, **g**, **j**, **m**, **p** UMAP visualization where cells are colored by cell type/labels/clusters. **b**, **e**, **h**, **k**, **n**, **q** Same UMAP visualization but cells are colored by cell bias score (CBS). **c**, **f**, **i**, **l**, **o**, **r** CBS distributions of cells from different cell types/batches/clusters. Boxes are colored by cell clusters using the same color palette as the first column. The centerline, bounds of box, top line, and bottom line of the boxplots represent the median, 25th to 75th percentile range, 25th percentile − 1.5 × interquartile range (IQR), and 75th percentile + 1.5 × IQR, respectively. The cell numbers for all boxplots are listed in Supplementary Dataset 4.

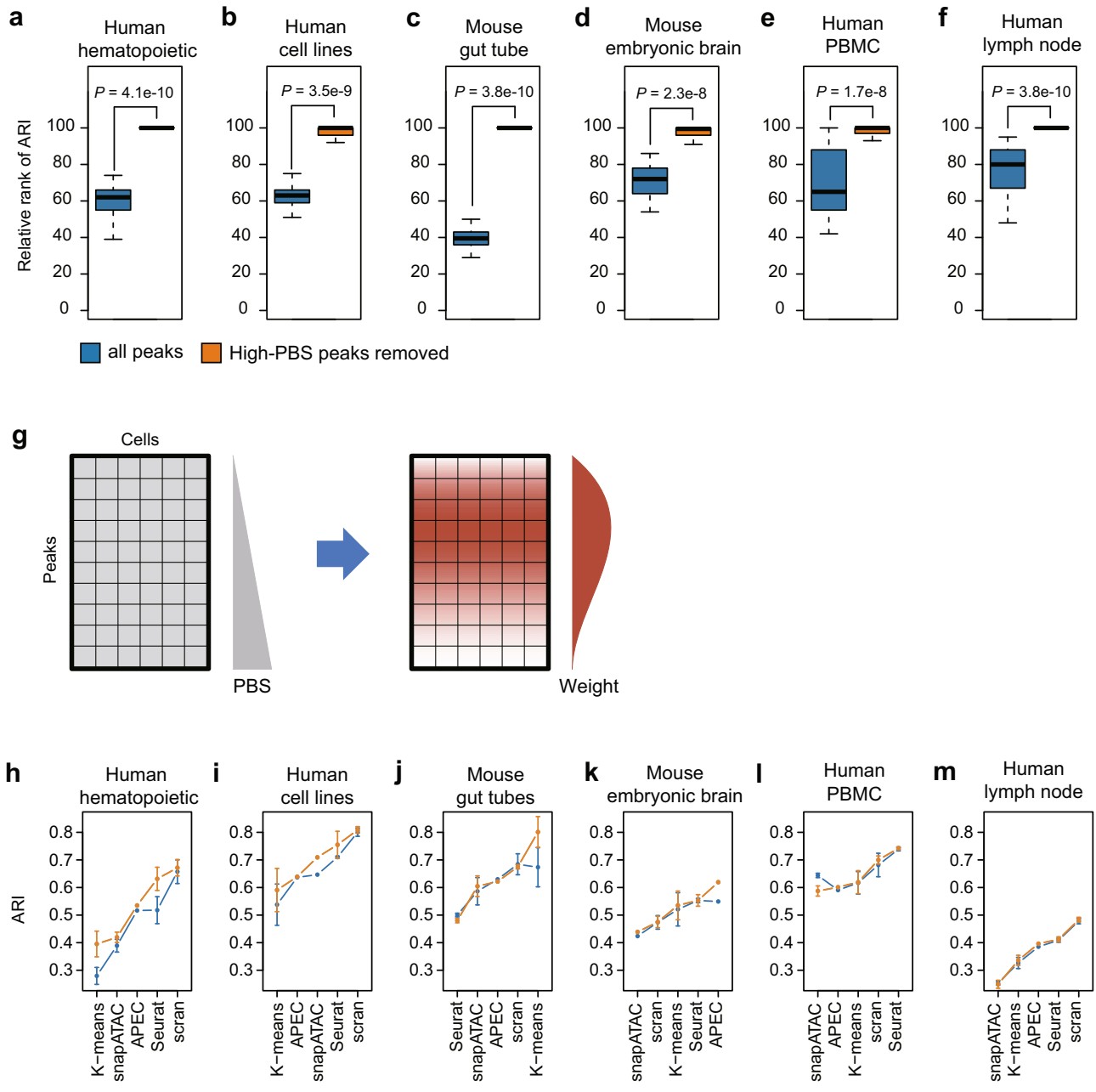

**Fig. 6 | SELMA bias correction model improves single-cell ATAC-seq cell clustering.** Comparisons of cell clustering accuracy before and after considering the peak bias score (PBS) on scATAC-seq data for human hematopoietic cells (**a**), mixed human cell lines (**b**), mouse primitive gut tube (**c**), and 10× Single-Cell Multiome data for mouse embryonic brain (**d**), human PBMC (**e**), and human lymph node (**f**). K-means clustering was performed after PCA dimensionality reduction. Blue, using all ATAC-seq peaks; orange, after removing 1–50% of peaks with the highest peak bias score (PBS). For each percentage of peaks retained from 50% to 99% with a 1% increment, 100 randomly sampled subsets of peaks were used as the background for determining the relative ranks of all peaks or retained peaks. The relative ranks of the adjusted Rand index (ARI), defined as the rank relative to the 100 randomly sampled sub-datasets, for the 50 cases from 50% to 99%, were plotted (*n* = 50 for each boxplot). *P* values were calculated by the one-sided Wilcoxon signed-rank test. The centerline, bounds of box, top line, and bottom line of the boxplots represent the median, 25th to 75th percentile range, 25th percentile – 1.5 × interquartile range (IQR), and 75th percentile + 1.5 × IQR, respectively. **g** Schematic of SELMA single-cell peak bias correction model. Peaks are weighted and adjusted based on PBS percentile using an empirically determined weight function. **h**–**m** Comparisons of the accuracy (measured by ARI) of single-cell clustering generated using different existing tools on scATAC-seq data with (orange) or without (blue) SELMA single-cell peak bias correction. Each panel represents the result for a scATAC-seq sample from a different biological system or experimental platform: human hematopoietic cells (**h**), mixed human cell lines (**i**), mouse gut tube (**j**), mouse embryonic brain (**k**), human PBMC (**l**) and human lymph node (**m**). Each data point at the center of the error bar represents the average (mean) ARI generated from 100 runs with different random seeds using the method labeled on the x-axis. The error bar represents the standard deviation (*n* = 100).

classification (Supplementary Fig. 11, "Methods"). We applied this weight function to adjust the peak-by-cell read count matrix for intrinsic bias correction and used the bias-corrected data matrix for cell clustering analysis. To evaluate the performance, we tested several commonly used single-cell data analysis tools, including APEC[40], Seurat[41], scran[42], and snapATAC[43], in addition to K-means, for scATAC-seq cell clustering, and compared the cell clustering accuracy of the uncorrected raw data with the bias-corrected data. While different tools showed various performances, across the 6 biological samples we tested, the bias-corrected data yielded a significantly higher ARI than the uncorrected data did for most cases (Fig. 6h–m, Supplementary Fig. 12a–f, Supplementary Dataset 5), including all cases for human hematopoietic cells (Fig. 6h, Supplementary Fig. 12a) and mixed human cell lines (Fig. 6i, Supplementary Fig. 12b), the two datasets with actual cell-type ground truth. While the absolute level of difference in ARI varies (Supplementary Fig. 12g), the overall improvement is statistically significant ($P = 0.03$), under the clustered one-sided paired t-test taking correlation structures within datasets into account[44]. Furthermore, regardless of which clustering method was used, the approach that yielded the highest ARI in each of the 6 samples is always using the bias-corrected data. These results demonstrated that SELMA can reduce the effect of intrinsic cleavage biases in scATAC-seq data and can be applied to existing single-cell data analytical tools to improve the accuracy of cell clustering analysis.

## Discussion

The existence of enzymatic cleavage biases in DNase-seq and ATAC-seq experiments has been widely acknowledged in the field, but to what extent such intrinsic biases affect data analysis, especially on the single-cell level, has not been systematically assessed. SELMA provides a quantitative approach for the accurate and robust estimation of intrinsic cleavage biases in both bulk and single-cell chromatin accessibility sequencing data and requires fewer sequence reads than the naive k-mer model. Taking Tn5 dimerization into consideration, SELMA yields more accurate bias estimation specifically for ATAC-seq data by including k-mer sequences at two Tn5 cleavages 9 bp apart. SELMA can improve functional analysis and interpretation of chromatin accessibility profiles. On the bulk level, we showed that considering SELMA-estimated biases can improve TF binding inference from ENCODE DNaseI consensus footprints for most TFs, with better performances compared to existing tools, including some that already considered "biases". On single-cell level, we showed widespread existence of intrinsic cleavage biases in single-cell ATAC-seq data, and demonstrated that SELMA single-cell bias reduction model can enhance the performance of existing tools and can increase the accuracy of cell type clustering. Therefore, SELMA can help researchers obtain more biological insights from chromatin accessibility data.

SELMA is built on top of the widely used k-mer model by combining simplex encoding and a linear model. Simplex encoding has the unique ability to capture the pairwise symmetry and orthogonality between mononucleotides and interactions within each dinucleotide. It significantly reduces the degrees of freedom without losing any variance information compared to the naive k-mer model. These properties enable SELMA to make robust estimations from fewer sequence reads or smaller datasets than are required by other approaches, hence enabling de novo bias estimation from a smaller DNA molecule, such as mtDNA in a DNase/ATAC-seq sample, without having to generate a separate genomic naked DNA sample. However, SELMA still relies on sufficient read counts for each k-mer for an accurate estimation. Although the performance of SELMA may increase with k, this effect is not unlimited. The performance using 12-mer is not as good as using 10-mer possibly because there are not enough reads in the dataset for many 12-mers. Nevertheless, SELMA works well for most existing DNase/ATAC-seq datasets tested in our study. In addition, the feasibility and superior performance of simplex encoding have also been demonstrated for TF motif characterization[45]. Following similar encoding strategies, SELMA can potentially be applied to any k-mer-based model or to any high-throughput sequencing data for robust sequence bias estimation and pattern recognition.

The k-mer biases estimated by SELMA are consistent across species and cell types, reflecting the assumption that the cleavage biases are intrinsic features of the enzymatic assays. Therefore, one can directly use the SELMA-estimated k-mer biases from DNaseI or Tn5-digested naked DNA data as universal intrinsic biases for DNase-seq and ATAC-seq, respectively, and incorporate these precalculated bias scores into the data analysis. However, this bias dataset is not guaranteed to remain accurate in many species that have not been profiled, as there might be unknown biases that have not been characterized. Although we are confident that the SELMA-estimated results should still be largely valid, one can always use the SELMA package for de novo estimation of cleavage biases from one's own datasets.

When applying SELMA-estimated FBS to correct biases in DNase footprints, our data were limited to ENCODE DNaseI consensus footprints as a proof-of-principle study. While this ENCODE dataset is thus far the largest DNase footprint repertoire, users might be interested in de novo detection of footprints from their customized DNase/ATAC-seq data. As many bioinformatics tools are already available for such tasks using various computational models[20], SELMA or SELMA-generated bias information can be incorporated into any of those models to improve the performance for more accurate regulatory DNA identification from footprints. On single-cell ATAC-seq analysis, while we show that SELMA single-cell bias correction model can achieve an overall significantly more accurate cell clustering on several publicly available datasets using a few existing tools, performance still varies across these tools and across different biological samples. One potential issue that limits a larger-scale benchmarking is the lack of ground truth for most existing scATAC-seq datasets. Except for the human hematopoietic cell sample and the mixed human cell line sample that have the known cell type labels as ground truth, we had to use transcriptome profiling scRNA-seq data as a "pseudo" ground truth or solver standard to determine the cell type identities, which not only depends on the quality of as sparse scRNA-seq data, but also implies the strong but debatable assumption that chromatin accessibility profiles should carry the same cell identity characterization as transcriptomic profiles. Such imperfect "pseudo" ground truths might have caused an insignificant improvement in many of the cases we tested. Meanwhile, the absolute level of increase in ARI also varies across clustering methods and datasets, and some are relatively small. One possible explanation is that ARI is a global metric considering all pairs of cells equally and exhaustively. In the real data, however, the intrinsic bias might only affect a small subset of cells, and the clustering accuracy would only be improved on that subset of cells, which results in a small ARI increase. Nevertheless, the overall increase in the cell clustering consistency for different biological systems tested indicates that SELMA bias correction reduces confounding noises in the data while biologically meaningful variances are retained[46]. In summary, SELMA is a universal and systematic bias reduction model and can be used to enhance the performance of existing methods and to improve single-cell chromatin accessibility profiling analysis.

## Methods

### High-throughput sequencing data collection and processing

Publicly available ATAC-seq, single-cell ATAC-seq, DNase-esq and ChIP-seq data used in this study were collected from Gene Expression Omnibus (GEO) and the ENCODE project. The metadata and accession numbers are listed in Supplementary Dataset 1.

Bulk ATAC-seq and DNase-seq data were processed as follows: Raw sequencing reads were aligned to the GRCh38 (hg38) reference genome with bowtie2 (v2.2.9) (-X 2000 for paired-end data)[47]. Low-quality reads (MAPQ < 30) were discarded. For paired-end sequencing data, reads with two ends aligned to different chromosomes (chimeric reads) were also discarded. For paired-end data, reads with identical 5′ end positions for both ends were regarded as redundant reads and discarded. The nonredundant reads were separated into chromosomal DNA (chromatin reads) and mitochondrial DNA (mtDNA reads) based on their genomic location. Peak detection was performed on the nonredundant chromatin reads using MACS2[48] (v2.1.2) (-q 0.01, --extsize 50) and ±200 bp centered on the peak summits was collected as the genome-wide chromatin accessible regions. The accessible regions in each dataset were separated into 5 groups from highest accessibility to lowest accessibility based on the read count on each peak (for Supplementary Fig. 1). The 5′ end nucleotides of each read were piled up to generate the genome-wide observed cleavage profile.

Single-cell ATAC-seq data were processed as follows: For the human hematopoietic cells and the mixed human cell line samples, raw sequencing reads were aligned to the GRCh38 (hg38) reference genome with bowtie2 (-X 2000). Low-quality reads (MAPQ < 30), chimeric reads and duplicate reads for each individual cell were discarded. For the mouse gut tube sample, scATAC-seq data from the 10x Genomics platform were preprocessed with Cell Ranger ATAC (v6.1.1) with the default parameters to generate fragments for each individual cell. For the 10x Single-Cell Multiome datasets, the processed fragment files were downloaded from the 10x genomics website (https://www.10xgenomics.com/resources/datasets/fresh-embryonic-e-18-mouse-brain-5-k-1-standard-2-0-0, https://www.10xgenomics.com/resources/datasets/pbmc-from-a-healthy-donor-no-cell-sorting-10-k-1-standard-2-0-0, https://www.10xgenomics.com/resources/datasets/fresh-frozen-lymph-node-with-b-cell-lymphoma-14-k-sorted-nuclei-1-standard-2-0-0). The fragment files were then used as input for the subsequent analysis. Because the Cell Ranger ATAC/ARC pipeline shifted from the Tn5 cleavage sites to +4/-5 bp in generating the fragment file, the coordinates were shifted back to represent the actual cleavage loci. For scATAC-seq datasets, cells with more than 10,000 reads were retained for analysis. For 10x Single-Cell Multiome datasets, cells pre-selected by Cell Ranger ARC and with more than 10,000 reads in both scRNA-seq and scATAC-seq parts were retained for analysis.

For transcription factor motif analysis, the sequence motifs of human TFs were collected from the HOCOMOCO database[36] (v11), and the genome-wide motif sites of TFs were detected by FIMO (v4.12.0) in the MEME package[49]. Motif sites located outside of the genome-wide 36 bp unique mappable regions were excluded from the analysis. In total, 61,531,309 motif sites for 156 TF motifs from the human genome were included in the analysis.

## SELMA model
In the naive k-mer bias model for intrinsic cleavage bias estimation, the naive k-mer biases were calculated as described in a previous study[13]. In brief, a naive k-mer bias was estimated as the observed frequency of the cleavages with the k-mer relative to the frequency of that k-mer in the background. For each mapped sequence read in a DNase-seq or ATAC-seq dataset, the enzymatic cleavage site was between genomic positions $i$ and $i$-1 for the plus (+) strand reads and between $i$ and $i$+1 for the minus (−) strand reads, where $i$ represents the genomic position of the 5′ nucleotide of the reads. The associated k-mer sequence was thus assigned as $[i - \frac{k}{2}, i + \frac{k}{2} - 1]$ for the plus strand reads and $[i - \frac{k}{2} + 1, i + \frac{k}{2}]$ for the minus strand reads. The naive k-mer bias score for k-mer $j$ is defined as the number of all observed cleavages with that k-mer

divided by the occurrences of that k-mer in the background:

$$S_j = \frac{N_j}{M_j} \tag{1}$$

where $N_j$ is the count of cleavages with k-mer $j$, and $M_j$ is the total count of occurrences of k-mer $j$ in the background (Fig. 1a). In the case of chromatin DNA, this background included 400 bp from each chromatin accessible region centered at the peak summit detected by MACS2. The background in the naked DNA samples included genome-wide 36 bp unique mappable regions. The background in mtDNA included the mitochondrial DNA sequence. The median was further subtracted from the bias scores for visualization in the scatter plots (e.g., Fig. 1d–k).

In the naive k-mer model, the bias score for each k-mer is independent and empirically determined from the data. The number of independent variables in the model is the total number of k-mers, i.e., $4^k$.

In SELMA, a simplex encoding model followed by a linear model was employed on top of the naive k-mer model to better estimate the intrinsic cleavage biases for each k-mer. To efficiently encode a k-mer sequence considering their intrinsic similarities, a simplex encoding model was adopted from previous studies[25,26], in which the DNA nucleotides were encoded as vectors representing the four tetrahedral vertices of a regular 0-centered simplex (Fig. 1b).

$$
\begin{aligned}
A &= [\ 1 \quad -1 \quad -1\ ] \\
C &= [-1 \quad 1 \quad -1\ ] \\
G &= [-1 \quad -1 \quad 1\ ] \\
T &= [\ 1 \quad 1 \quad 1\ ]
\end{aligned} \tag{2}
$$

In the simplex encoding, the vectors representing the four nucleotides are of equal length, mutually orthogonal, and equidistant from each other. To account for interactions between adjacent nucleotides, a dinucleotide was additionally encoded as the outer product of the two vectors associated with the two nucleotides:

$$
\begin{aligned}
AA &= [\ 1 \quad -1 \quad -1 \quad -1 \quad 1 \quad 1 \quad -1 \quad 1 \quad 1\ ] \\
AC &= [-1 \quad 1 \quad -1 \quad 1 \quad -1 \quad 1 \quad 1 \quad -1 \quad 1\ ] \\
AG &= [-1 \quad -1 \quad 1 \quad 1 \quad 1 \quad -1 \quad 1 \quad 1 \quad -1\ ] \\
AT &= [\ 1 \quad 1 \quad 1 \quad -1 \quad -1 \quad -1 \quad -1 \quad -1 \quad -1\ ] \\
CA &= [-1 \quad 1 \quad 1 \quad 1 \quad -1 \quad -1 \quad -1 \quad 1 \quad 1\ ] \\
CC &= [\ 1 \quad -1 \quad 1 \quad -1 \quad 1 \quad -1 \quad 1 \quad -1 \quad 1\ ] \\
CG &= [\ 1 \quad 1 \quad -1 \quad -1 \quad -1 \quad 1 \quad 1 \quad 1 \quad -1\ ] \\
CT &= [-1 \quad -1 \quad -1 \quad 1 \quad 1 \quad 1 \quad -1 \quad -1 \quad -1\ ] \\
GA &= [-1 \quad 1 \quad 1 \quad -1 \quad 1 \quad 1 \quad 1 \quad -1 \quad 1\ ] \\
GC &= [\ 1 \quad -1 \quad 1 \quad 1 \quad -1 \quad 1 \quad -1 \quad 1 \quad -1\ ] \\
GG &= [\ 1 \quad 1 \quad -1 \quad 1 \quad 1 \quad -1 \quad -1 \quad -1 \quad 1\ ] \\
GT &= [-1 \quad -1 \quad -1 \quad -1 \quad -1 \quad -1 \quad 1 \quad 1 \quad 1\ ] \\
TA &= [\ 1 \quad -1 \quad -1 \quad 1 \quad -1 \quad -1 \quad 1 \quad -1 \quad -1\ ] \\
TC &= [-1 \quad 1 \quad -1 \quad -1 \quad 1 \quad -1 \quad -1 \quad 1 \quad -1\ ] \\
TG &= [-1 \quad -1 \quad 1 \quad -1 \quad -1 \quad 1 \quad -1 \quad -1 \quad 1\ ] \\
TT &= [\ 1 \quad 1 \quad 1 \quad 1 \quad 1 \quad 1 \quad 1 \quad 1 \quad 1\ ]
\end{aligned} \tag{3}
$$

One can show that these vectors for dinucleotide interactions are also of equal length, mutually orthogonal, and equidistant from each other. In fact, in the simplex encoding, mononucleotides and dinucleotides were encoded as rows in a Hadamard matrix of order 4 and a Hadamard matrix of order 16, respectively.

Considering both mononucleotides and dinucleotide interactions, a k-mer can then be encoded as $k$ mononucleotides and $k - 1$ dinucleotides, plus an intercept term. Therefore, the dimensionality of

a k-mer simplex encoding is

$$p(k) = 1 + 3k + 9(k - 1) = 12k - 8 \quad (4)$$

In SELMA, the intrinsic k-mer biases were then estimated in a linear model with these $12k - 8$ parameters using the observed naive k-mer bias scores (Fig. 1c). In detail, we have

$$y \sim \sum_{i=1}^{12k-8} a_i x_i \quad (5)$$

where each observation $y$ is the naive k-mer bias score; $x_i \in \{1, -1\}$, is the independent variable in the simplex encoding vectors; and $a_i$ is the parameter to be estimated. After linear regression, the model-fitted value $\hat{y}$ was defined as the SELMA bias score for each k-mer.

For the genome-wide cross-correlation analysis presented in Fig. 2b, c, reads from plus (+) and minus (−) strands on chromatin accessible regions (peaks) were collected separately to generate plus strand observed cleavage profile and minus strand observed cleavage profile, respectively. The Pearson correlation coefficient between the plus strand signal and the k-bp shifted minus strand signal is:

$$\rho_k = \frac{\sum_i (P_i - \bar{P})(M_{ik} - \overline{M_k})}{\sqrt{\sum_i (P_i - \bar{P})^2 \sum_i (M_{ik} - \overline{M_k})^2}} \quad (6)$$

where $P_i$ is the log-scaled plus strand cleavage count at genomic position $i$ (log2(cleavage+1)), $M_{ik}$ is the log-scaled minus strand cleavage count at genomic position $i$ with a k-bp shift, $\bar{P}$ is the mean of all the $P_i$, and $\overline{M_k}$ is the mean of $M_{ik}$ for all $i$. All genomic positions on the genome-wide DNase/ATAC-seq peaks were included in the analysis. $k$ is chosen from 1 to 20 (x-axis in Fig. 2b, c).

Different bias estimation methods were implemented for comparison as follows: We use $a_i^+$ and $a_i^-$ to denote the "5′ only" intrinsic sequence bias score at genomic position $i$ on the plus strand and minus strand, respectively. We use $a'^+_i$ and $a'^-_i$ to denote the bias score from other bias estimation methods in this section. Different bias estimation methods used in Fig. 2 and its associated sections are listed below:

- For the "5′ only" method, $a_i^+$ was calculated based on the k-mer ratio associated with the k-mer spanning positions $[i - \frac{k}{2}, i + \frac{k}{2} - 1]$ on the plus strand, and $a_i^-$ was calculated in a similar way based on the nucleotides spanning genomic positions $[i - \frac{k}{2} + 1, i + \frac{k}{2}]$ on the minus strand (reverse complement of the DNA sequence on the corresponding plus strand). This method was applied to both DNase-seq and ATAC-seq, while the other methods were applied only to ATAC-seq as they were specifically designed for ATAC-seq.
- For SELMA (Fig. 2a), the bias score was calculated as the geometric mean of the "5′ only" bias score at the given position and the "5′ only" bias score at 9 bp downstream of the other strand, i.e., $a'^+_i = \sqrt{a_i^+ \times a_{i+9}^-}$, and $a'^-_i = \sqrt{a_i^- \times a_{i-9}^+}$
- For the model in Martins et al[14], the bias score at genomic position $i$ was calculated in a similar way to the "5′ only" method but using a gapped 11-mer model. The model could be represented as $XXXXXXNNXNXCXXNNXNNNNXNX$, in which position $i$ was represented by $C$; positions that were ignored were represented by $X$ and informative positions were represented by $N$.
- For the model in Baek et al[18], the bias score at genomic position $i$ was calculated in a similar way to the "5′ only" method but the cleavages were shifted +4/−5 bp for +/− strand cleavages. In practice, following the description in the "bagfoot" package, the

bias score at position $i$ was calculated as $a'^+_i = a_{i+5}^+$, and $a'^-_i = a_{i-5}^+$.

- For the model in Calviello et al.[8], the bias score at genomic position $i$ was calculated in a similar way to the "5′ only" method but using the 6-mer bias table provided in the study.

Observed and bias-expected cleavages were calculated as follows: The 1 bp at 5′ end positions for DNase-seq or ATAC-seq reads were piled up to generate the genome-wide observed cleavage profile. To generate the bias-expected cleavage profile, for a 50-bp window centered on nucleotide $i$, we placed the same number of observed cleavages following the sequence bias contribution in this window. Let $\widehat{N_i^s}$ represent the bias-expected cleavage at position $i$ on strand $s \in \{+, -\}$, $N_i^s$ represent the observed cleavage at position $i$ on strand $s$, and $a_i^s$ denote the intrinsic cleavage bias (estimated with any of the above methods) at position $i$ on strand $s$. Then we have

$$\widehat{N_i^s} = N_i^s y_i^s \quad (7)$$

where

$$y_i^s = \frac{2^{a_i^s}}{\sum_{j=i-25}^{i+24} 2^{a_j^s}} \quad (8)$$

We used the Pearson correlation coefficient to compare the observed cleavage profile and the bias-predicted cleavage profile (Fig. 2). We only considered positions within the accessible regions to ensure that all positions had sufficient reads in the 50-bp window for accurate estimation.

## DNaseI footprint analysis

The genome-wide DNaseI consensus footprint regions were downloaded from Ref. 7 (https://resources.altius.org/~jvierstra/projects/footprinting.2020/consensus.index/). A total of 4,460,438 footprint regions that do not contain unidentified nucleotides (N) in the GRC38 (hg38) reference genome were used for subsequent analyses. The observed DNaseI cleavage profile from a DNase-seq dataset and DNaseI SELMA bias scores across ±50 bp centered on the footprint region were plotted as heatmaps (Fig. 4a, b). The footprint regions were ordered by the footprint lengths, and each 1000 footprint regions with similar lengths were compressed as one row in the heatmap for better visualization. The plus- and minus-strand signals were plotted separately. We aligned the footprint regions based on the two observed bias spikes in each footprint region (located 7 bp to the right of the left boundary and 7 bp to the left of the right boundary of the footprint, labeled as -0 and +0 in Fig. 4c–h). The center regions between the bias spikes were scaled to 4 bins to align footprint regions with different lengths.

The footprint bias score (FBS, in Fig. 4i–k) was defined as the difference between "spike bias" and "center bias". The "spike bias" was calculated as the average of the two SELMA bias scores at the spike positions, while the "center bias" was calculated as the median SELMA bias score at the rest of the positions in the footprint region. Let $FBS_j$, $b_j$, and $c_j$ denote the footprint bias score, spike bias and center bias of footprint $j$, respectively. The FBS, spike bias, and center bias are given by:

$$FBS_j = b_j - c_j \quad (9)$$

$$b_j = \overline{a_{i \in B}^{s \in \{+, -\}}} \quad (10)$$

$$c_j = \widetilde{a_{i \in C}^{s \in \{+, -\}}} \quad (11)$$

where $a_i^s$ represents the SELMA bias score at genomic position $i$ on the strand $s \in \{+, -\}$, overbar represents the average, tilde represents the median, and $B$ and $C$ represent spike positions (within the flanking 1 bp of the bias spikes) and the remaining positions of the footprint $j$, respectively. To calculate the randomly shuffled FBS (random k-mer bias), the FBS was calculated in the same way, but the bias score $a_i^s$ for each position was randomly selected from the SELMA bias score table.

The following methods were used to calculate footprint scores for comparison:

- Raw footprint: The raw footprint score was calculated following a previous study[13], using the formula $= -\left(\log \frac{n_C+1}{n_R+1} + \log \frac{n_C+1}{n_L+1}\right)$, where $n_C$, $n_R$ and $n_L$ denote the DNase cleavage count in the motif region, and the flanking regions to the right and left of the motif, respectively. The flanks are both the same length as the motif.
- Wellington footprint: We used Wellington[9] (v0.2.0) with default parameters to detect genome-wide footprints and selected significant footprint regions with $p$ values < 1e−10. The output footprint score was assigned to the TF motif overlapping with the footprint region.
- HINT footprint: We used HINT[11,19] (v0.12.3) with default parameters to detect genome-wide footprints. For bias correction mode, we used an additional parameter: --bias-correction. The footprint score of each footprint region was assigned to the overlapping TF motifs.

Inferences of TF binding with different features were implemented as follows: For each TF in Fig. 4j, the TF motifs overlapping with consensus footprint regions were collected as target regions. DNase-seq read count in the footprint region ("reads"), footprint score from an existing method, and bias score were used as features in a multinomial logistic regression model to infer TF occupancy at footprint-overlapping motif regions. For each available method, model training with cross-validation and predictions were performed using different combination of features: "original method" refers to reads + footprint score. An additional feature of either SELMA FBS or a random k-mer bias was added for different models. We used a performance measure integration approach[11] to evaluate different models' prediction power. For each model, we calculated the area under the ROC curve (AUROC) at 100%, 10%, and 1% false-positive rate (FPR). We also calculated the area under the precision-recall curve (AUPRC) at 100%, 10%, and 1% recall. We then combined these six performance measures as a rank score $S$, defined as

$$S = \frac{1}{6} \sum_i -\log \frac{r_i}{N+1} \tag{12}$$

where $i = 1, \ldots, 6$ denotes the 6 performance measures, $r_i$ is the rank of a model for performance measure $i$, and $N$ is the total number of models.

To calculate the random k-mer bias, we randomly permuted the SELMA k-mer bias table and generated the k-mer bias table for a "simulated enzyme". We used this "simulated" bias table to calculate the FBS for each footprint region and performed TF inference modeling similar to what we did for the DNaseI SELMA FBS. This permutation was performed 100 times to generate 100 performance rank scores for random k-mer bias used as controls.

Footprint prediction power of TF binding on motif sites was assessed as follows: In Fig. 4j–k, we collected the genome-wide motif sites overlapping with consensus footprint regions and the ChIP-seq peaks for each TF with HOCOMOCO[36] motif and ChIP-seq data available in human cell lines. We collected data from all human cell lines with both DNase-seq and more than 20 TF ChIP-seq samples available from ENCODE, resulting in 7 cell lines, 375 TF ChIP-seq samples, and 156 TFs (Supplementary Dataset 1). In total, we surveyed genome-wide motif sites for 156 TFs, and 61,531,309 motif sites in total. In Fig. 4k, for

each TF, we selected two subgroups of its motif sites based on the FBS of their overlapped footprint regions: the top 10% of motif sites with the highest FBS, defined as "sites with high-bias footprint"; and the bottom 10% of motif sites with the lowest FBS, defined as "sites with low-bias footprint". We calculated the proportion of motif sites overlapping with TF ChIP-seq peaks for each of the two subgroups and plotted on a scatter plot (Fig. 4k). To test whether low-bias footprints tend to have more TF binding than high-bias footprints for most TFs, we conducted a t-test comparing the distribution of TF binding log likelihood ratios of low-bias over high-bias footprints to the standard normal distribution, and the test p-value is labeled in Fig. 4k.

## Single-cell ATAC-seq clustering analysis

For single-cell ATAC-seq data in the human hematopoietic cells and human cell line samples, the cell-type information for each individual cell was used as the ground truth, or the gold standard. For single-cell ATAC-seq data in the mouse gut tube sample, the cell-type information was assigned based on label transfer[40] from the single-cell RNA-seq dataset in the same system[48] (GSE136689), as the "pseudo" ground truth, or the silver standard. In detail, we integrated scRNA-seq and scATAC-seq data using the ArchR package[50] (v1.0.1). Individual cells with a high RNA integration score (unconstrained predicted score < 0.56) were collected as high-quality cells for the analysis. The cutoff of the RNA integration score (0.56) was determined by maximizing the interclass variance in the RNA integration score using Otsu's method[51]. Cell types represented by fewer than 10 cells were discarded. For the 10x Single-Cell Multiome datasets, scRNA-seq parts from each sample were separated and clustered using Seurat (v4.0)[41] with default parameters. The scRNA-seq clustering results were used as the "pseudo" ground truth for scATAC-seq cell clusters.

The average bias for each individual cell (median SELMA cell bias score, median CBS, used in Fig. 5) was calculated as the median of the bias for all the fragments in the individual cell. The bias for each paired-end fragment was calculated as the mean of the SELMA bias scores for the 5′ end and the 3′ end.

Peak detection and peak bias score calculation were performed as follows: We first combined all the single-cell ATAC-seq reads in the dataset and performed MACS2[48] peak calling with a q-value cutoff of 0.1 to include all the potential accessible regions in the human hematopoietic cell and mixed human cell line samples (100,456 peaks detected for human hematopoietic cells and 83,318 peaks for mixed human cell lines, respectively). For the 10x Genomics single-cell data, the $q$ value cutoff of 0.01 was used for peak calling (146,098 peaks detected for mouse gut tube; 52,086 peaks for mouse embryonic brain; 83,491 peaks for human PBMC; and 78,243 peaks for human lymph node, respectively). Potential accessible regions with fewer than 10 covered reads or more than 4000 covered reads were discarded in the subsequent analysis. To consider the effect of intrinsic cleavage bias in scATAC-seq data, we summarized the bias for each potential accessible region by calculating the peak bias score (PBS), defined as the median SELMA bias score of all reads (from all individual cells) located in the peak region.

The SELMA single-cell peak bias correction model was designed as follows: The goal of this model is to give a different weight to all scATAC-seq peaks based on the PBS, so that peaks heavily affected by intrinsic biases have a lower weight while peaks less affected by intrinsic biases have a higher weight when they are used to perform cell clustering analysis. To estimate the relative contribution of each peak in a scATAC-seq dataset to the clustering result, we conducted an analysis of variance (ANOVA) for each identified peak, using the scATAC-seq read count/signal across different cells as the variable and the known cell-type labels as the target group labels. The F score from the ANOVA output was used to quantify the contribution of each peak to the clustering. Within each percentile of peaks based on their PBS ranks, the median F score of the peaks with ANOVA $p$ value < 0.05 was used to represent the relative level of contribution for this percentile

of peaks (Supplementary Fig. 11a–f). Based on the contribution patterns of the scATAC-seq samples tested, a standard beta distribution was used to model the percentile weight function:

$$W(x) = \frac{x^{\alpha-1}(1-x)^{\beta-1}}{B(\alpha, \beta)} \quad (13)$$

where $x = 0, 0.01, 0.02,..., 0.99$ is the percentile of the peak ranked by PBS; $\alpha, \beta > 1$ are the shape parameters; $B(\alpha, \beta)$ is the beta function:

$$B(\alpha, \beta) = \frac{\Gamma(\alpha)\Gamma(\beta)}{\Gamma(\alpha + \beta)} \quad (14)$$

where $\Gamma()$ is the gamma function. The beta distribution was chosen because its probability density function has the following properties: (1) constraints: $W(0) = 0$ and $W(1) = 0$; (2) normalization: $\int_0^1 W(x)dx = 1$; (3) asymmetry with mode at $x = \frac{\alpha-1}{\alpha+\beta-2}$. Based on the relative contribution pattern in Supplementary Fig. 11a–f, the parameters were empirically determined as $\alpha = 2, \beta = 3$. Therefore, the weight function becomes

$$W(x) = 12x(1-x)^2 \quad (15)$$

After the read count in each peak in each cell was weighted using this weight function, the whole read count matrix was scaled back to keep the total read count in the matrix unchanged from the raw data matrix. The adjusted read count matrix then underwent the clustering analysis. For the single-cell analysis tools that require raw scATAC-seq reads as input, including Scran and Seurat in the ArchR package and snapATAC, the adjusted number of reads for each peak in each cell was synthesized from each peak region and assigned to the cell.

scATAC-seq cell clustering and evaluation were performed as follows: For k-means clustering as a naive method, we performed PCA dimensionality reduction on the accessible regions by individual cells matrix of normalized read count and kept 60 PCs, followed by k-means clustering. The number of clusters (k) in k-means clustering was determined as the actual number of cell types in the dataset. For published methods tested, including APEC[40], Seurat[41], scran[42], and snapATAC[43], we applied each method with default parameters. To evaluate the accuracy of cell clustering, we used the adjusted Rand index (ARI)[39] between a clustering result with the predefined cell-type labels (either the actual cell-type label as the ground truth or the scRNA-seq projected cell cluster label as the pseudo ground truth). To evaluate the robustness of clustering results for each method, we repeated the clustering for 100 times with different random seeds, and used the average and standard deviation of the ARIs from the 100 runs as the evaluation metrics. APEC result is invariant with random seeds so 100 repeats were not applied to APEC. We selected the top 60 dimensions for those methods at the dimensional reduction step (PCA, Seurat, scran and snapATAC). We used the ArchR package[50] to implement the Seurat and scran clustering methods. To explore the effect of intrinsic cleavage bias on scATAC-seq analysis, we selected different percentages (from 50% to 99%, with a 1% increment) of peaks with the lowest PBS (i.e., removing 50%-1% of peaks with the highest PBS) as input to perform cell clustering (Fig. 6a–f). We also randomly selected the same percentage of peaks as a control to estimate the relative rank of the clustering performance from using all peaks and using retained peaks. In detail, for each percentage from 50% to 99%, we randomly sampled peaks 100 times and defined the relative rank as the number of ARIs from random samples that were less than the ARI from the same percentage of retained peaks or all peaks. The relative ranks for different percentages from 50% to 99% are summarized and plotted as boxplots in Fig. 6a–f. To evaluate

the improvement in clustering accuracy after SELMA single-cell peak bias correction, for each method applied to each dataset, we compared the ARIs from the 100 runs with different random seeds between corrected and uncorrected data, and used the one-sided Wilcoxon rank-sum test to assess the statistical significance of clustering accuracy improvement (Supplementary Fig. 12a–f). To assess the statistical significance of the overall improvement after bias correction, we compared all the average ARIs from each method and each sample between uncorrected and corrected data, using the one-tailed paired clustered t-test, taking correlation structures into account[44]. In practice, we performed the test using the ttestClust function in the R package htestClust[52], with scATAC-seq sample labeled as groups. We also performed a clustered Wilcoxon signed-rank test using the wilcoxtestClust function; both tests produced a similar $P$ value of 0.032.

### Reporting summary

Further information on research design is available in the Nature Research Reporting Summary linked to this article.

## Data availability

The mouse gut tube scATAC-seq dataset is available in the Gene Expression Omnibus (GEO) with accession number GSE168373. The 10x Single-Cell Multiome datasets are downloaded from 10x Genomics website (https://www.10xgenomics.com/resources/datasets/fresh-embryonic-e-18-mouse-brain-5-k-1-standard-2-0-0, https://www.10xgenomics.com/resources/datasets/pbmc-from-a-healthy-donor-no-cell-sorting-10-k-1-standard-2-0-0, https://www.10xgenomics.com/resources/datasets/fresh-frozen-lymph-node-with-b-cell-lymphoma-14-k-sorted-nuclei-1-standard-2-0-0). All publicly available data used in this study are downloaded from the GEO or the ENCODE project data portal. Accession numbers for all the GEO and ENCODE data used in the study are available in Supplementary Dataset 1.

## Code availability

The SELMA package is available at Github at https://github.com/zang-lab/SELMA and Zenodo at https://doi.org/10.5281/zenodo.7048767[53]. User instructions and example data files can be found in the README document. Essential annotation data, analysis results, and scripts are also available in the repository.

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

## Acknowledgements

The authors thank Dr. Xiaole Shirley Liu for helpful discussions and Dr. Tae-Hee Kim for sharing the mouse primitive gut tube scATAC-seq data.

This work was supported by US National Institutes of Health (NIH) grants R35GM133712 and K22CA204439 (to C.Z.), R35GM128635 (to M.J.G.), National Science Foundation grant NSF-2048991 and University of Pittsburgh Center for Research Computing (to T.Z.), UVA Cancer Center Farrow Fellowship and the NCI Cancer Center Support Grant P30 CA44579 (to S.S.H.).

## Author contributions

C.Z. conceived and directed the research. S.S.H., C.A.M., and C.Z. designed the method. L.L., M.J.G., K.D., and T.Z. contributed to the method design. S.S.H. implemented the method and analyzed the data. L.L., Q.L., and W.M. contributed to the data analysis and interpretation. S.S.H. and C.Z. wrote the manuscript. All authors approved the manuscript.

## Competing interests

The authors declare no competing interests.
