## [Peer Review File · Nature Communications]

Reviewers' Comments:

Reviewer #1:

Remarks to the Author:

In this manuscript, Hu and colleagues reported a new computational tool (SELMA) to evaluate the intrinsic cleavage biases from genomic chromatin accessibility profiling datasets. They have shown that this method can be helpful for TF enrichment inference and scATAC-seq clustering analysis. It is well known that Tn5 and DNase I present intrinsic cleavage bias that may influence the accuracy of open chromatin region detection. In this regard, previous efforts (as mentioned in this study) have been made for bias correction. Although the authors have provided evidence that SELMA outperforms existing methods, there is no result showing that this effect is biologically relevant. For example, can this method help identify additional accessible elements/TFs that are important for key biological process? Or can this method identify cell types that could be missed by other methods? If yes experimental validations are also needed for any new claim.

Besides, here are few comments/questions:

1. All supplementary figures are missing, which makes it difficult to evaluate the robustness of this method.
2. In line 111-112, why it is $1+3k+9(k-1)$, rather than $4k+9(k-1)$?
3. In line 261, the authors claim that considering intrinsic cleavage bias could improve the performance of TF binding inference. However, only 48 TFs were used for the comparison, more TFs (there are hundreds in the database) should be included.
4. Different family TFs have distinct motifs, the author should provide a comprehensive analysis to show how many TFs can be affected when doing enrichment analysis because of intrinsic cleavage biases?
5. For cluster analysis, why drop 50% high bias data, instead of using corrected data?

Reviewer #2:

Remarks to the Author:

In this work, Shengen Shawn Hu and colleagues presented a novel method, SELMA, to estimate the intrinsic cutting biases in DNase-seq and ATAC-seq data, and showed that the biases can be used to improve downstream analysis including in the single-cell ATAC-seq analysis. Different from existing methods, SELMA built upon k-mer model with a more efficient simplex encoding therefore it can adapt much bigger k (e.g. >10 , compared with normal 6-mer in other methods). The authors found that the biases estimated from SELMA are consistent across species and cell types, and additionally SELMA can estimate de novo biases from data from mtDNA. While applying the biases on downstream analysis, the authors claimed that the TF prediction at DNase-seq/ATAC-seq footprints and the single-cell ATAC-seq clustering can be improved. Overall, the method is attractive and solid and the manuscript is well-written. However, I have the following comments regarding the benefits of applying the biases estimated from SELMA:

1. In Figure 4j, while predicting TF binding sites from chromatin accessibility, the variance of the rank score is much larger while integrating SELMA FBS (footprint biases scores) than original and original+random k-mer biases (the orange boxes in Fig 4j). It indicates that integrating SELMA FBS may not always improve the TF prediction. The same clue can be found while the authors evaluated the effect of using intrinsic biases for different TFs. They found that the binding sites of some TFs have higher chances to appear in high FBS (high biases) regions (Fig 4k). They suspected that these TFs may have stronger DNA-binding affinities than others. However, this raises a critical concern since the SELMA FBS can not be universally beneficial for TF binding sites prediction, and it's not clear when SELMA FBS should be applied for what TFs with how 'strong' the affinities. Please explore and suggest when the biases should be considered and when shouldn't.
2. In Figure 5, the authors investigated how the SELMA biases can improve clustering in single-cell ATAC-seq. When the authors generated sets of 'debiased' peaks, they wrote 'SELMA-debiased peaks (excluding 1%-50% of peaks with the highest CBS)' in line 309, but 'we selected different percentages (from 50% to 99%, with a 1% increment) of peaks with the lowest PBS (SELMA debiased peaks...)' in line 658. 'CBS' is defined as the cell biases score and 'PBS' is defined as the 'peak biases score'. So which one is correct? Also, the results in Fig 5d-j are confusing. First, it

seems that for all the comparisons, except for Fig 5j for snapATAC clustering, even removing 1% of peaks with top biases can greatly improve the relative rank of ARI. Please confirm whether this is true. Additionally, why the improvement in the snapATAC method is so insignificant?

Minor:

1. In Figure 4k, are the p-values from two-tails or one-tail t-test? Also, it's hard to draw the conclusion that (line 611) the low bias footprints tend to have more TF binding than high-bias footprints from this test. It depends on the TFs included in this study. One can only conclude that among the TFs included in the study, there are more TFs that will be improved with SELMA FBS than what won't be improved with SELMA.

2. In the method for TF motif analysis, why (line 424, 443) 36bps is specifically chosen to determine the genome-wide unique mappable regions? In other studies, people found that the mappability depends on the read length of the sequencing. (<https://www.frontiersin.org/articles/10.3389/fgene.2014.00381/full>) As for the current technologies, 36bps size is too small. Or in another word, in this study, there may be more regions excluded while using the 36bps mappability profile.

Point-by-point Response to Reviewers' comments

Reviewer #1 (Expertise: experimental, chromatin organization):

In this manuscript, Hu and colleagues reported a new computational tool (SELMA) to evaluate the intrinsic cleavage biases from genomic chromatin accessibility profiling datasets. They have shown that this method can be helpful for TF enrichment inference and scATAC-seq clustering analysis. It is well known that Tn5 and DNase I present intrinsic cleavage bias that may influence the accuracy of open chromatin region detection. In this regard, previous efforts (as mentioned in this study) have been made for bias correction. Although the authors have provided evidence that SELMA outperforms existing methods, there is no result showing that this effect is biologically relevant. For example, can this method help identify additional accessible elements/TFs that are important for key biological process? Or can this method identify cell types that could be missed by other methods? If yes experimental validations are also needed for any new claim.

Response: We thank the reviewer for acknowledging the evidence that SELMA outperforms existing methods. Meanwhile, we have shown the biological relevance of the intrinsic biases and the effect of bias correction in several aspects: 1) DNase footprint analysis considering SELMA-estimated biases can yield more accurate TF binding inference genome-wide, as a global improvement, as shown in Fig. 4j. In the revised manuscript, we have tested DNase footprint analysis in 7 human cell lines covering 156 different TFs, which are all the TFs with ChIP-seq data available from ENCODE in cell lines that also have DNase-seq data (Fig. S7a and Response Fig. 1 below). 2) SELMA bias correction can help generate more accurate cell clustering from single-cell ATAC-seq data. As cell clustering is essential for characterizing cell types and understanding cell identity; a more accurate clustering can produce more biologically meaningful interpretations. Because the actual cell type information as ground truth is frequently lacking in single-cell data, we used the only 2 publicly available single-cell ATAC-seq datasets with known cell type labels, plus 4 additional datasets using single-cell RNA-seq inferred cell type classification as silver standards, to demonstrate that SELMA does result in more accurate cell clustering results, as shown in Fig. 6. We believe these data have shown the biological relevance of intrinsic bias correction using SELMA.

Besides, here are few comments/questions:

1. All supplementary figures are missing, which makes it difficult to evaluate the robustness of this method.

Response: Supplementary figures were submitted in the manuscript submission system but were not successfully linked to the version that was sent to reviewers. We have updated the Supplementary materials including supplementary figures and supplementary tables in the revised manuscript.

2. In line 111-112, why it is $1+3k+9(k-1)$, rather than $4k+9(k-1)$?

Response: As described in the Methods section (lines 515-518) and illustrated in Fig. 1b, in the SELMA model, each mononucleotide is encoded as a 3-dimensional vector; each dinucleotide is additionally encoded as the outer product of the two associated mononucleotides, i.e., additional 9 dimensions. Therefore, a k -mer encoding model has $3k$ parameters for k mononucleotides, $9(k-1)$ parameters for $k-1$ dinucleotides, and 1 additional parameter for the constant term/intercept in the model. The total number of parameters is $1+3k+9(k-1)$.

Corresponding revision: Lines 113, 114, 516.

3. In line 261, the authors claim that considering intrinsic cleavage bias could improve the performance of TF binding inference. However, only 48 TFs were used for the comparison, more TFs (there are hundreds in the database) should be included.

Response: TF binding patterns and chromatin accessibility profiles are cell-type-specific. To evaluate the performance of TF binding inference from DNase footprints, one has to use the DNase-seq data and TF ChIP-seq data from the same cell type. In the revised manuscript, we have analyzed *all* human cell lines that have both DNase-seq data and more than 20 ChIP-seq data samples available in the ENCODE database. This includes 375 ChIP-seq

samples for 156 different TFs in 7 cell lines (Table S1). These 156 TFs are *all* TFs that have ChIP-seq data in a cell type where DNase-seq data is also available from ENCODE. As shown below (Response Fig. 1) and Fig. S7a in the revised manuscript, adding more cell lines will not further increase the total number of TFs included. We have updated the results from this expanded analysis showing the TF binding inference improvement after considering the SELMA footprint bias (Figs. 4j, S7, Table S2). As a result, for most TFs, considering SELMA footprint bias can improve the prediction power of DNase-seq footprint to TF binding. Such improvement is consistently observed when using different footprint detection methods.

Response Figure 1 (Figure S7a): Number of transcription factors (TFs) covered vs. number of cell lines included in the analysis. Cell lines are ranked by the number of ChIP-seq samples available.

Corresponding revision: Figures 4j, 4k; S7; Tables S1, S2. Lines 260–279.

4. Different family TFs have distinct motifs, the author should provide a comprehensive analysis to show how many TFs can be affected when doing enrichment analysis because of intrinsic cleavage biases?

Response: We have included a comprehensive analysis in the revised manuscript to address this question. Among the 375 TF ChIP-seq samples we have included in the updated analysis, 323 (86%) show an improvement of binding inference from DNase footprints when considering the SELMA footprint bias for at least one method. This involves 139 TFs (89%) out of the total of 156 TFs included. Depending on which specific method

was used for footprint score calculation, considering SELMA footprint bias can improve the inference performance for 277-291 (74-78%) ChIP-seq samples, covering 114-117 (73-75%) TFs. We show the results for all TFs samples in Figure S8 and Table S2, which include the differential rank score measuring the inference performance for each TF sample. The summary of the differential rank score of all TF samples for each of the 4 footprint methods tested is presented in Figures 4j & S7b-h.

Corresponding revision: Figures 4j; S7, S8; Table S2; Lines 265–279.

5. For cluster analysis, why drop 50% high bias data, instead of using corrected data?

Response: To clarify, in the original manuscript, bias correction was done by dropping a certain percentage of data with the highest bias. We showed that in most cases, dropping any percentage (1%-50%) can result in an improved clustering result (Figure S10). In this case, “using corrected data” means the exact same thing as “dropping high bias data”.

In the revised manuscript, to overcome the arbitrariness of the high-bias-peak-removal approach, we provide a new approach with a bias correction model for single-cell ATAC-seq cell clustering analysis. The new model gives a weight to every peak region instead of dropping any data and can result in a significant improvement in clustering accuracy in all three datasets from different biological systems (Figure 6g-m, S11, Table S4).

Corresponding revision: Figure 6g-m, S10, S11, Table S4, Lines 303–309, 323–344.

Reviewer #2 (Expertise: compbio, chromatin organization):

In this work, Shengen Shawn Hu and colleagues presented a novel method, SELMA, to estimate the intrinsic cutting biases in DNase-seq and ATAC-seq data, and showed that the biases can be used to improve downstream analysis including in the single-cell ATAC-seq analysis. Different from existing methods, SELMA built upon k-mer model with a more efficient simplex encoding therefore it can adapt much bigger k (e.g. >10, compared with normal 6-mer in other methods). The authors found that the biases estimated from SELMA are consistent

across species and cell types, and additionally SELMA can estimate de novo biases from data from mtDNA. While applying the biases on downstream analysis, the authors claimed that the TF prediction at DNase-seq/ATAC-seq footprints and the single-cell ATAC-seq clustering can be improved. Overall, the method is attractive and solid and the manuscript is well-written. However, I have the following comments regarding the benefits of applying the biases estimated from SELMA:

1. In Figure 4j, while predicting TF binding sites from chromatin accessibility, the variance of the rank score is much larger while integrating SELMA FBS (footprint biases scores) than original and original+random k-mer biases (the orange boxes in Fig 4j). It indicates that integrating SELMA FBS may not always improve the TF prediction. The same clue can be found while the authors evaluated the effect of using intrinsic biases for different TFs. They found that the binding sites of some TFs have higher chances to appear in high FBS (high biases) regions (Fig 4k). They suspected that these TFs may have stronger DNA-binding affinities than others. However, this raises a critical concern since the SELMA FBS can not be universally beneficial for TF binding sites prediction, and it's not clear when SELMA FBS should be applied for what TFs with how 'strong' the affinities. Please explore and suggest when the biases should be considered and when shouldn't.

Response: We agree with the reviewer that the larger variance of the rank score after integrating SELMA FBS may lead to a confusing interpretation of the TF prediction improvement. Instead, we have now calculated and presented the difference in the rank score for each TF comparing before and after considering SELMA FBS, to directly illustrate the prediction improvement by considering the SELMA FBS for each individual TF binding profile (Figure 4j, delta rank score). We have also updated the number of TF binding profiles (ChIP-seq samples) included in the analysis to 375 (including 156 TFs in 7 cell lines, Table S1) to increase the coverage of the analysis. We found that 323 (86%) samples show improvement after considering SELMA FBS from at least one method (Figure S7, Figure S8, Table S2). For example, Zinc finger family TFs including CTCF and REST showed the highest improvement after considering footprint bias, while the SOX family (e.g., SOX5) and HLH family (e.g., MYC) TFs rarely showed improved inference performance after considering footprint bias, consistent with previous studies about the association between footprint strength and residence time of the TF on DNA. These data provide a reference for users to determine whether the biases should be considered when studying certain TFs.

Corresponding revision: Figures 4j, S7, S8; Tables S1, S2; Lines 260–279.

2. In Figure 5, the authors investigated how the SELMA biases can improve clustering in single-cell ATAC-seq. When the authors generated sets of 'debiased' peaks, they wrote 'SELMA-debiased peaks (excluding 1%-50% of peaks with the highest CBS)' in line 309, but 'we selected different percentages (from 50% to 99%, with a 1% increment) of peaks with the lowest PBS (SELMA debiased peaks...' in line 658. 'CBS' is defined as the cell biases score and 'PBS' is defined as the 'peak biases score'. So which one is correct?

Response: We thank the reviewer for catching the typo in line 309. It should be 'PBS'. We have corrected it in the revised manuscript (now in Line 324).

Also, the results in Fig 5d-j are confusing. First, it seems that for all the comparisons, except for Fig 5j for snapATAC clustering, even removing 1% of peaks with top biases can greatly improve the relative rank of ARI. Please confirm whether this is true. Additionally, why the improvement in the snapATAC method is so insignificant?

Response: The reviewer's interpretation is correct. To ensure the clarity of the data presentation, we plot the percentage difference of ARI before and after removing 1% to 50% of high bias peaks for the 6 scATAC-seq datasets included, shown below (Response Fig. 2) and as Figure S10 in the revised manuscript. In some cases, removing 1% of high-bias peaks (keeping 99% peaks) indeed increases the ARI, but not for every dataset. To overcome the arbitrariness of the high-bias peak-removal approach, in the revised manuscript, we provide a new approach with a bias correction model for single-cell ATAC-seq cell clustering analysis. The new model gives a weight to each peak instead of removing any peak and can result in improvement of clustering accuracy for all 6 datasets including the 3 from the original manuscript and 3 newly added 10x Single Cell Multiome datasets, and for every method tested, including snapATAC (Figure 6g-m, Table S4).

Corresponding revision: Figures 6, S10; Table S4; Lines 323–344.

Response Figure 2 (Figure S10): Percentage difference of ARI of single cell clustering after removing 1-50% of scATAC-seq peaks with the highest PBS compared to using all peaks. Each panel represents a different scATAC-seq dataset tested. In each panel, x-axis represents the percentage of peaks retained for clustering; y-axis represents percentage difference of ARI between the K-means clustering result and the ground truth cell labels.

Minor:

1. In Figure 4k, are the p-values from two-tails or one-tail t-test?

Response: They are one-tail t-tests. We have added this information in the figure legend.

Corresponding revision: Figure 4 legends

Also, it's hard to draw the conclusion that (line 611) the low bias footprints tend to have more TF binding than high-bias footprints from this test. It depends on the TFs included in this study. One can only conclude that among the TFs included in the study, there are more TFs that will be improved with SELMA FBS than what won't be improved with SELMA.

Response: We agree with the reviewer and have modified the conclusion as suggested.

Corresponding revision: Lines 286–288.

2. In the method for TF motif analysis, why (line 424, 443) 36bps is specifically chosen to determine the genome-wide unique mappable regions? In other studies, people found that the mappability depends on the read length of the sequencing.

(<https://www.frontiersin.org/articles/10.3389/fgene.2014.00381/full>) As for the current technologies, 36bps size is too small. Or in another word, in this study, there may be more regions excluded while using the 36bps mappability profile.

Response: Following the reviewer's comment, we repeated the analysis using 50bp and 100bp mappable regions, respectively, and found that the results are consistent with those from 36bp mappable regions (see Response Fig. 3 below for the same analysis as in Figure 4j), suggesting that the conclusions are robust to read length mappability. Therefore, we keep the original data in the manuscript.

Response Figure 3: Differential rank score comparing performance of TF binding inference models similar to Figure 4j. Analysis were done in motif-containing DNase footprints in the uniquely mappable regions of short sequence reads of different lengths. From top to bottom, 36bp, 50bp, and 100bp mappable regions, respectively.

Reviewers' Comments:

Reviewer #1:

Remarks to the Author:

The authors have addressed all my concerns. I don't have additional comments and think it's now suitable for publication. Congratulations for the nice work.

Reviewer #2:

Remarks to the Author:

Thanks for addressing my previous comments and I am satisfied with most of the answers and edits to the manuscript. However, the newly added results, for evaluating the scATAC-seq data, raised my questions on how significant the approach to considering cutting biases can improve single-cell clustering.

The authors found that removing highly biased peaks may not increase clustering accuracy for every dataset (6 datasets, figure S10 and response figure 2), so they proposed a new model to put weight on each peak. Although they admitted so in the address to the reviewer -- 'In some cases, removing 1% of high-bias peaks (keeping 99% peaks) indeed increases the ARI, but not for every dataset', the main text in the manuscript is still: 'For all 6 scATAC-seq datasets, cell clustering after removing 1%-50% of peaks with the highest PBS largely increased ARI from using all peaks (Fig. S10).' We can clearly see that the improvement can range from significant (figure S10a, from 0 to 50%) to marginal (figure S10e, from minus to +3%), from stable improvement (figure S10a) to inconsistent improvement (figure S10b and S10e).

When applying the weighted model for clustering, we do see some improvement in ARI however the improvements shown in figure 6h-m are somehow trivial. First of all, the y-ranges of figure 6h-m are all different. Please use the same range, for example, 0.25 to 0.8. Figure S6m, human lymph node, shows almost ignorable improvement and the overall ARI values are low indicating the 'ground truth' of clustering borrowed from the scRNA-seq analysis may not be correct. The snapATAC result in figure 6I, human PBMC, even shows a decrease in performance (contradicting the 'always reached the highest ARI' claim). Therefore, please consider using some measurements to evaluate the overall improvement (for example the distribution of the difference in ARI).

RESPONSE TO REVIEWER COMMENTS

Reviewer #1 (Remarks to the Author):

The authors have addressed all my concerns. I don't have additional comments and think it's now suitable for publication. Congratulations for the nice work.

Response: We thank the reviewer for the positive feedback.

Reviewer #2 (Remarks to the Author):

Thanks for addressing my previous comments and I am satisfied with most of the answers and edits to the manuscript. However, the newly added results, for evaluating the scATAC-seq data, raised my questions on how significant the approach to considering cutting biases can improve single-cell clustering.

The authors found that removing highly biased peaks may not increase clustering accuracy for every dataset (6 datasets, figure S10 and response figure 2), so they proposed a new model to put weight on each peak. Although they admitted so in the address to the reviewer -- 'In some cases, removing 1% of high-bias peaks (keeping 99% peaks) indeed increases the ARI, but not for every dataset', the main text in the manuscript is still: 'For all 6 scATAC-seq datasets, cell clustering after removing 1%-50% of peaks with the highest PBS largely increased ARI from using all peaks (Fig. S10).' We can clearly see that the improvement can range from significant (figure S10a, from 0 to 50%) to marginal (figure S10e, from minus to +3%), from stable improvement (figure S10a) to inconsistent improvement (figure S10b and S10e).

Response: We agree with the reviewer's comment and have revised the main text for clarifications. Now the corresponding sentences have been revised as: "For most cases in the 6 scATAC-seq datasets, cell clustering after removing 1%-50% of peaks with the highest PBS can increase ARI from using all peaks (Fig. S10). Although the level of improvements varies across different datasets and not every percentage of peak removal yields a higher ARI, such improvement in clustering accuracy by removing high-bias peaks is statistically

significant compared to the baseline of randomly removing the same number of peaks (Fig. 6a-f).” (lines 323–328).

When applying the weighted model for clustering, we do see some improvement in ARI however the improvements shown in figure 6h-m are somehow trivial. First of all, the y-ranges of figure 6h-m are all different. Please use the same y-range, for example, 0.25 to 0.8.

Response: The ranges of y-axis in Figures 6h-6m have now been made identical, following the reviewer’s comment. The revised figure can be found below:

Figure 6h-m: Comparisons of the accuracy (measured by ARI) of single-cell clustering generated using different existing tools on scATAC-seq data with (orange) or without (blue) SELMA single-cell peak bias correction. Each panel represents the result for an scATAC-seq sample from a different biological system or experimental platform: human hematopoietic cells (h), human cell lines (i), mouse gut tube (j), mouse embryonic brain (k), human PBMC (l) and human lymph node (m). Each dot in each panel represents the average ARI for a different method labeled on the x-axis. Error bars represent the standard deviations of ARIs generated from 100 runs with different random seeds.

Figure S6m, human lymph node, shows almost ignorable improvement and the overall ARI values are low indicating the 'ground truth' of clustering borrowed from the scRNA-seq analysis may not be correct. The snapATAC result in figure 6l, human PBMC, even shows a decrease in performance (contradicting the 'always reached the highest ARI' claim). Therefore, please

consider using some measurements to evaluate the overall improvement (for example the distribution of the difference in ARI).

Response: Following the reviewer’s suggestions, we added a few statistical measurements to evaluate the overall effect of SELMA single-cell bias correction on scATAC-seq cell clustering.

First, 4 of the 5 clustering methods we tested (K-means, Seurat, scran, and snapATAC) can generate different results given different initialization seeds. Therefore, for each of these 4 methods on each of the 6 scATAC-seq datasets, we repeated the analysis for 100 times using different random seeds, and generated 100 ARIs for both uncorrected and SELMA-corrected data for comparison. As shown in Supplementary Figure S12a-f (below), 15 out of the 24 cases show a significantly increased ARI after bias correction, under the p-value threshold of 0.05. Another 2 cases have p-values between 0.05 and 0.1. It is worth noting that for Human hematopoietic cells (a) and mixed human cell line samples (b), which are the only two scATAC-seq datasets with “actual” ground truth, the increase in ARI after SELMA bias correction was consistently significant for all methods. Insignificant improvement only occurred in datasets that have imperfect “pseudo” ground truths derived from scRNA-seq information.

Supplementary Figure S12(a-f): Distributions of ARIs generated from 100 runs with different random seeds, compared between uncorrected raw data (blue) and SELMA-corrected data (orange), for each clustering method in each scATAC-seq sample. P-values were calculated by the one-sided Wilcoxon rank sum test. Different panels represent data from different biological systems: human hematopoietic cells (a), human cell lines (b), mouse gut tubes (c), mouse embryonic brain (d), human PBMC (e), and human lymph node (f).

We subsequently revised Figures 6h-6m by showing the average and standard deviation (as error bars) of ARI from 100 runs with different random seeds whenever applicable.

Second, we followed the reviewer’s suggestion and analyzed the distribution of the difference in ARIs from all 5 methods for all 6 datasets. While the absolute level of difference varies, the vast majority of the cases (25/30) do show a higher ARI after SELMA bias correction (Supplementary Figure S12g, below). We then performed a clustered one-sided paired t-test to compare the ARIs between corrected and uncorrected data, taking correlation structures within dataset into account, and obtained a p-value of 0.03195. Alternatively, the clustered Wilcoxon rank sum test also produced a significant p-value of 0.03183. These results indicate the significance of overall improvement in clustering accuracy by using SELMA bias correction.

Supplementary Figure S12g: Difference in ARI after bias correction. HSC: human hematopoietic cells, CL: human cell lines, Gut: mouse gut tubes, Brain: mouse embryonic brain, PBMC: human PBMC, Lymph: human lymph node.

Third, our data indicate that different tools have different clustering performances measured by ARI. For each scATAC-seq dataset, we want to compare different tools, with or without SELMA bias correction, and will call the one that yields the highest ARI as the best performing approach among the 10 approaches (5 clustering methods x 2 bias considerations). As a result, for every dataset, the approach that yields the highest ARI is always one of those with SELMA bias correction. This is what we meant by claiming SELMA-corrected data “always reached the highest ARI.” We made some clarifications about this in the revised manuscript (lines 340–348).

Finally, while showing the significance of the overall improvement, we agree that the level of improvement (effect size) varies across clustering methods and datasets, and in some cases can be fairly small. There are several potential reasons for this. Firstly, as the reviewer pointed out and mentioned above, the “pseudo ground truth” inferred from scRNA-seq might contain errors. They are not “true” ground truth. Unfortunately, with existing data that are available, this is the best we can do. As we stated in the Discussion section, the lack of ground truth for most existing scATAC-seq data limits us from doing a larger-scale benchmarking study. However, we would like to emphasize that for the only two scATAC-seq datasets with “true” ground truth (Human hematopoietic cells and Human mixed cell line samples), the improvement in clustering accuracy after SELMA bias correction was both significant ($P = 0.03$ for each dataset, by one-sided paired Wilcoxon rank sum test). Secondly, ARI is a global metric considering all pairs of cells equally and exhaustively. In the real data, however, the intrinsic bias might only affect a subset of cells (Figure 5), and the clustering accuracy would only be improved on the subset of cells, which could result in a relatively small ARI increase. Nevertheless, we believe that our data is convincing to demonstrate the significance of clustering improvement after bias correction. We have revised the Discussion section accordingly (lines 407–410, 413–423).

Reviewers' Comments:

Reviewer #2:

Remarks to the Author:

Thanks for addressing my comments! No further concerns. The new figures are nice to describe the results!

RESPONSE TO REVIEWER COMMENTS

Reviewer #2 (Remarks to the Author):

Thanks for addressing my comments! No further concerns. The new figures are nice to describe the results!

Response: We thank the reviewer for the feedback.